# Reward biases spontaneous neural reactivation during sleep

Virginie Sterpenich [1,2,3 ✉], Mojca K. M. van Schie[1,3,4], Maximilien Catsiyannis[1], Avinash Ramyead[5], Stephen Perrig[6], Hee-Deok Yang[7], Dimitri Van De Ville [3,8,9] & Sophie Schwartz[1,2,3]

Sleep favors the reactivation and consolidation of newly acquired memories. Yet, how our brain selects the noteworthy information to be reprocessed during sleep remains largely unknown. From an evolutionary perspective, individuals must retain information that promotes survival, such as avoiding dangers, finding food, or obtaining praise or money. Here, we test whether neural representations of rewarded (compared to non-rewarded) events have priority for reactivation during sleep. Using functional MRI and a brain decoding approach, we show that patterns of brain activity observed during waking behavior spontaneously reemerge during slow-wave sleep. Critically, we report a privileged reactivation of neural patterns previously associated with a rewarded task (i.e., winning at a complex game). Moreover, during sleep, activity in task-related brain regions correlates with better subsequent memory performance. Our study uncovers a neural mechanism whereby rewarded life experiences are preferentially replayed and consolidated while we sleep.

[1] Department of Neuroscience, Faculty of Medicine, University of Geneva, Geneva, Switzerland. [2] Swiss Center for Affective Sciences, University of Geneva, Geneva, Switzerland. [3] Geneva Neuroscience Center, University of Geneva, Geneva, Switzerland. [4] Leiden University Medical Center, Leiden, Netherlands. [5] Department of Psychiatry, Weill Institute for Neurosciences, University of California, San Francisco, CA, USA. [6] Center of Sleep Medicine, Division of Pneumology, University Hospital Geneva, Geneva, Switzerland. [7] Department of Computer Engineering, Chosun University, Seosuk-dong, Dong-ku, Gwangju, Korea. [8] Department of Radiology and Medical Informatics, University of Geneva, Geneva, Switzerland. [9] Institute of Bioengineering, Ecole Polytechnique Fédérale de Lausanne, Lausanne, Switzerland. ✉email: Virginie.Sterpenich@unige.ch

Sleep contributes to memory consolidation[1]. Human neuroimaging findings suggest that brain regions activated during the encoding of waking experiences may be reactivated during subsequent non-rapid eye movement (NREM) sleep[2–4] and are associated with increased local slow-wave activity (SWA)[5,6]. These two mechanisms index parallel forms of memory optimization, i.e., neural replay and synaptic homeostasis[7,8], predominating during slow oscillatory activity and resulting in improved subsequent behavioral performance. Moreover, it is possible to influence memory reactivation during NREM by presenting sensory cues, for instance tones or odors that have previously been associated with specific stimuli or events[9], thus presumably enhancing hippocampal playback of those cued items[10]. These recent observations raise a fundamental question for memory research: how does the brain select those memories that will be reprocessed during sleep? As for an individual's survival, events with an affective relevance, namely those yielding highly positive (e.g., rewards) or negative (e.g., threats, punishments) outcomes, should have an elevated storage priority so as to optimize future behavior[11]. We would thus expect that emotionally-relevant memories have an advantage for reactivation processes occurring during sleep. Initial converging support for this hypothesis comes the following observations: (i) memory for both aversive and rewarding information benefits from sleep[12–16], (ii) emotional and reward networks (including the amygdala and ventral striatum) are activated during human sleep[17], and (iii) after a place-reward association task, hippocampal–striatal neuronal ensembles display coordinated replay during sleep in rats[18]. However, prior work did not test for whether and how the reactivation of neural activity corresponding to a rewarded event may compete with that of an equivalent, but nonrewarded event. Yet, although critical for current models of memory, direct experimental evidence supporting the privileged reemergence during sleep of neural activity corresponding to information with a high motivational or emotional value is still lacking. In this work, we analyze fMRI data acquired during task execution at wake and during subsequent sleep, and demonstrate that the reactivation of distributed patterns of neural activity could be detected during human sleep and predominates during slow-wave-rich sleep. Critically, by also showing that patterns of neural activity coding for information with high motivational relevance are prioritized in the competition for reactivation during sleep, the present study extends previous work, while uncovering a possible mechanism whereby rewarded memories benefit from sleep[12–14].

## Results

**Brain decoding approach to revealing reactivation in sleep.** Our main goal was to test whether brain patterns associated with a rewarded experience during wakefulness are prioritized for spontaneous reactivation during sleep. To this end, we used a brain decoding approach that allows the classification of mental states on the basis of measured brain states[4]. A pioneering study by Horikawa et al. (2013)[19] demonstrated that this approach is suitable for predicting the content of ongoing mental imagery during sleep based on fMRI activity. Here, we recorded simultaneous EEG-fMRI (64 channels EEG BrainAmps system; 3 T MRI scanner, Trio TIM, Siemens, Germany) in 18 participants (12 women, mean age ± SD: 22.1 ± 2.4 years) while they played a "face" game and a "maze" game, and while they subsequently slept (Fig. 1 and Supplementary Information). These two games were designed to recruit distinct and well-characterized brain networks specialized in the processing of face information (face game) and in spatial navigation (maze game). To investigate effects of reward on neural reactivation, we manipulated the

games so that, at the very end of the session, each participant randomly won either the face or the maze game. This randomization was used to ensure that any game-related reactivation during sleep was due to the reward status of one game (i.e., won or not), and not because spontaneous patterns of brain activity during sleep intrinsically resembling those elicited when participants played one of the two games (i.e., face or maze). A pattern classifier was trained on the fMRI data collected during game playing, and then applied to the sleep data. We could thus test whether the brain state associated with the victorious game (compared to the state associated with the non-rewarded game) prevailed during subsequent sleep, particularly during stage N3 sleep when slow oscillations predominate.

While awake, each participant played the two games during 60-s blocks organized in a pseudorandom order. Each game block was followed by a 90-s block of rest. In the face game, participants had to discover a target face among 18 different faces (Fig. 1a). Each face block started with a written clue relating to the target face. However, only the two last clues, nearing the end of the session, were truly informative about the target face. In the maze game, participants had to find the exit in a virtual maze (adapted from the Duke Nukem 3D game). On each maze block, one additional arrow indicating the way to the exit was disclosed. Mirroring the structure of the face game, the exit was only accessible at the very end of the maze game. As soon as the participant found the solution for one game (i.e., discovering the target face or the exit of the maze), the remaining blocks of the other game were rigged to make it unsolvable. Thus, each participant won either the face or the maze game. The difficulty of each task was adjusted during pilot experiments to ensure that participants were fully engaged in the task, and did not realize that the resolution of the games was manipulated, so that they attributed their victory in one game to their own good performance. Eight participants found the correct face and 10 participants reached the exit of the maze. As expected, each game recruited a specific set of distributed brain regions, including the fusiform face area (FFA) and occipital face area (OFA) for the face game, and the parahippocampal place area (PPA) for the maze game (Fig. 1b and Supplementary Table 1). After the game session and a short pause outside the scanner, the participants were invited to sleep in the scanner, while continuous EEG-fMRI data was recorded (sleep session; Fig. 1b). Because neural replay has previously been associated with slow oscillations[1], we included in our analyses the data from those 13 participants who reached sustained N3 sleep stage (mean age ± SD: 22.0 ± 2.5 years; 7 won the face and 6 the maze game; Supplementary Table 2). After a short debriefing on sleep quality and mentation during the sleep session, all participants left the lab to complete their night of sleep at home. After one supplemental recovery night, all participants came back to the lab where their memory for each game was tested.

**Reward-related fMRI reactivation in N3 sleep.** To assess the spontaneous reemergence during sleep of patterns of brain activity associated with the rewarded (i.e. won) and non-rewarded game at wake, we used a neural decoding approach (see Supplementary Information). We first extracted the time course of activity from regions of interest (ROIs) activated during each game, and during the blocks of Rest (Supplementary Table 1). Next, we trained a classifier to dissociate between 5 brain states. Two states corresponded to playing the face or maze game, subsequently assigned to the Reward (R) and No-Reward (NR) states according to which game each participant won and used for decoding during sleep. One state represented activity during the blocks of rest (Fig. 1b). We also included two states of no interest,

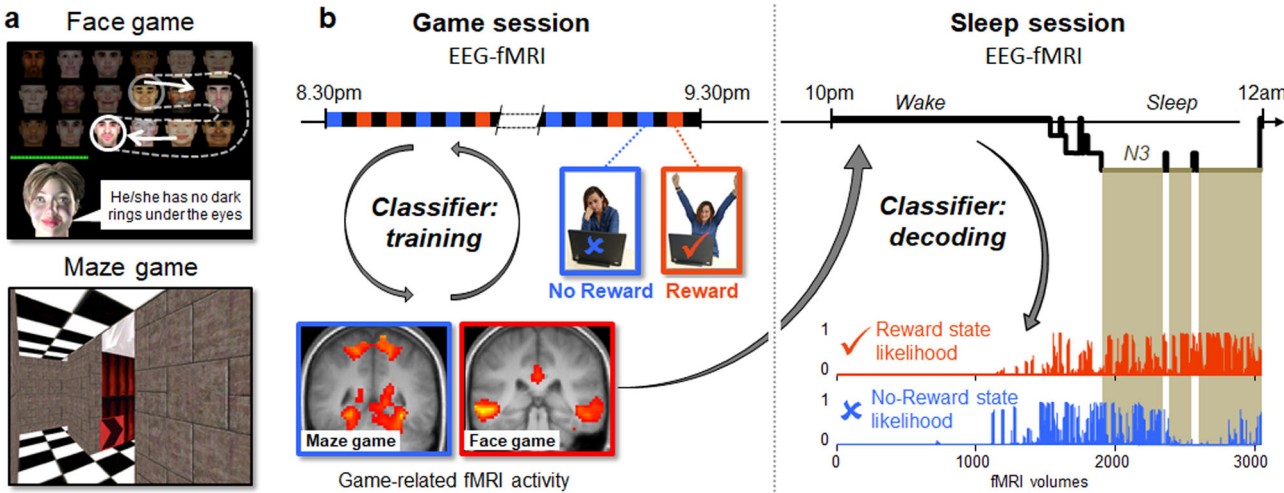

**Fig. 1 Experimental overview. a** In the face game (top), participants had to discover a target face based on a series of clues. For illustrative purposes, all faces are visible here, while in the actual game they were hidden, except for a small disk corresponding to the "torch", which participants moved using a trackball. In the maze game (bottom), participants had to find the exit of the maze, aided by arrows. **b** During the game session, participants played blocks of the face (in red) and maze (in blue) games, separated by blocks of rest (in black). During the last block, each participant won one of the games (here the face game is the rewarded game). After a 15-min break, participants underwent the sleep session. EEG-fMRI was recorded during both sessions. Each game recruited specific brain regions (blue frame: maze vs. face game; red frame: face vs. maze game) and the classifier was trained on these data to differentiate between different brain states (5 states in total, see Supplementary Information). Sleep EEG was scored and the classifier was applied to the fMRI from the sleep session to determine the likelihood for each brain state to occur at each fMRI scan during different sleep stages (example for the Reward and No-Reward states shown on the bottom right). N3 indicates N3 sleep stage.

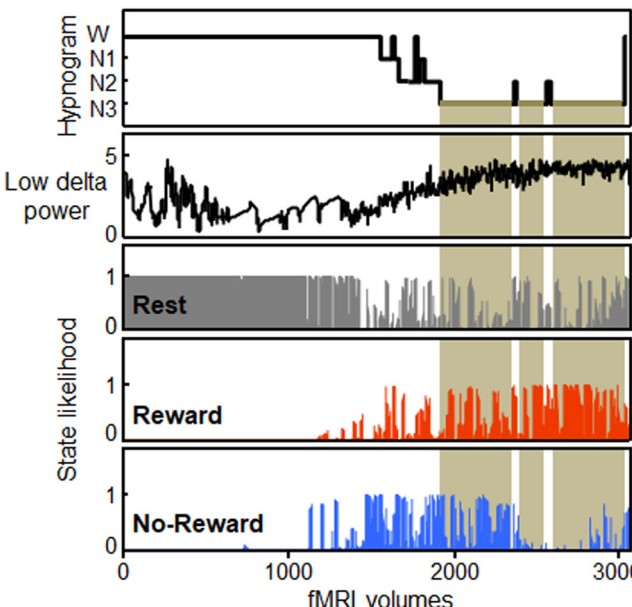

**Fig. 2 Data from one representative participant.** From top to bottom: Hypnogram, time course of the log-transformed power values for low delta (1–2 Hz), time-courses of likelihood for the Rest (in gray), Reward (in red), and No-Reward (in blue) brain states (pre-R and pre-NR are not represented here). Periods of N3 are highlighted in light brown. Increased likelihood of the brain state corresponding to the rewarded game is visible during N3. W indicates Wake; N1, N2, N3 indicate N1, N2, N3 sleep stages respectively.

pre-reward (pre-R) and pre-no-reward (pre-NR), corresponding to the 3 s preceding each game; see Supplementary Information. Successful classification of the main states of interest was validated using a leave-one-out procedure with the data from the game session (Supplementary Fig. 1). We then applied this classifier to the data from the sleep session, which provided, for each

acquired fMRI brain volume, the likelihood of each brain state to be present (from 0 to 1). Finally, after artifacts removal from the EEG data, we performed sleep staging and computed the average likelihood for each brain state to be reactivated during the distinct sleep stages in each participant. In other words, we could thus test whether the brain states associated to the rewarded and the non-rewarded game were differentially distributed across sleep stages (Figs. 1b and 2).

Specifically, a repeated-measure ANOVA on likelihood measures with Brain state (Reward, No-Reward, Rest, pre-R, pre-NR) and Sleep stage (Wake, N1, N2, N3) as within-subjects factors showed main effects of Brain state ($F_{(4,48)} = 137.1$, $p < 0.001$) and an interaction between Brain state and Sleep stage ($F_{(12,144)} = 14.8$, $p < 0.001$) (Fig. 3a, Supplementary Table 3). Post-hoc planned comparisons first showed that the Rest state predominated over the Reward and No-Reward states during wakefulness (Rest vs. Reward: $F_{(1,12)} = 263.2$, $p < 0.001$, Rest vs. No-Reward: $F_{(1,12)} = 515.8$, $p < 0.001$), while it was progressively less represented from light to deeper sleep stages (main effect of sleep for the Rest state: $F_{(1,12)} = 371.9$, $p < 0.001$). These results suggest a similarity between resting blocks during the game session and resting while awake during the sleep session. Conversely, both game-related brain states had a relatively low likelihood of being present from wakefulness to N2 (less than 0.20), and did not differ between Reward versus No-Reward states (Reward vs. No-Reward for Wake: $F_{(1,12)} = 2.24$, $p = 0.16$, N1: $F_{(1,12)} = 0.18$, $p = 0.67$, N2: $F_{(1,12)} = 0.03$, $p = 0.87$). By contrast, and confirming our main hypothesis, the Reward state has a much higher likelihood to occur during N3 sleep and differed from the No-Reward state ($F_{(1,12)} = 10.65$, $p = 0.007$). The states corresponding to the preparation period before each game did not differ for the Reward and No-Reward states in any sleep stage (pre-R vs. pre-NR states for Wake: $F_{(1,12)} = 0.002$, $p = 0.96$, N1: $F_{(1,12)} = 0.37$, $p = 0.56$, N2: $F_{(1,12)} = 0.03$, $p = 0.88$, N3: $F_{(1,12)} = 1.42$, $p = 0.26$). Note that the classifier was trained and applied using 5 possible distinct states, and that there was no correlation between the Reward and No-Reward states during N3 (Spearman

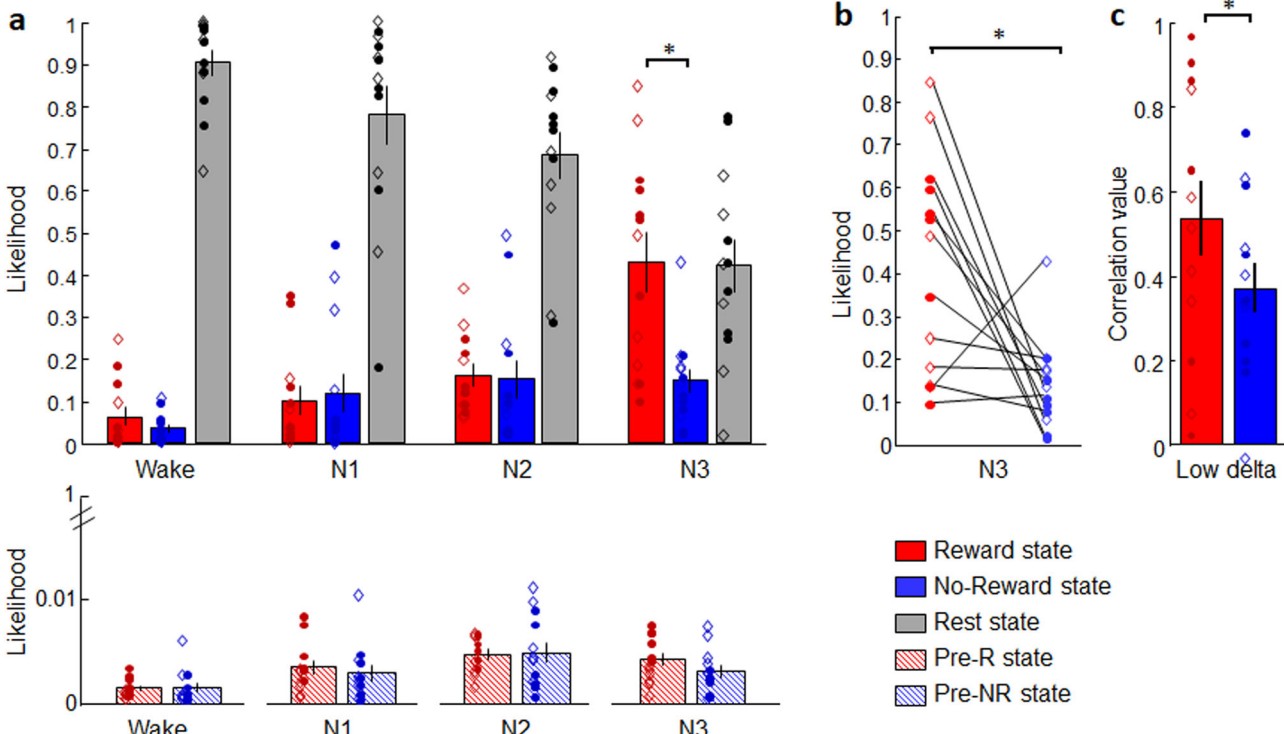

**Fig. 3 Classification results (N = 13). a** Mean likelihood of Reward, No-Reward, Rest, pre-Reward (pre-R) and pre-No-Reward (pre-NR) brain states in each sleep stage, showing increased reactivation of the brain state associated with the rewarded game during N3. Asterisk represents $p = 0.007$ for repeated measures ANOVA with post-hoc $t$ test (two-sided). **b** Individual data for the Reward and No-Reward state during N3 sleep. Asterisk represents $p = 0.007$ for $t$ test (two-sided). **c** Correlation between time course of likelihood and low delta power (1–2 Hz) for the Reward and No-Reward states. Error bars represent SEM (Standard Error of the Mean); dots represent individual values for those participants won the face game and diamonds correspond to data from participants who won the maze game. Asterisk represents $p = 0.03$ for t-test (two-sided). W indicates Wake; N1, N2, N3 indicate N1, N2, N3 sleep stages respectively. Source data are provided as a Source Data file.

rho $= -0.47$, $p = 0.14$), thus indicating that the presence of one of the two game-related states did not imply absence of the other state. To ensure that the type of game won (Face or Maze) did not affect the main results, we computed a new ANOVA with Brain state and Sleep stage as within-subjects factor and Game Won as between-subjects factor. We observed no main effect of Game Won ($F_{(1,11)} = 0.25$, $p = 0.63$), no interaction between Game Won and Brain state ($F_{(4,44)} = 0.27$, $p = 0.89$) and interaction between Game Won and Sleep stages ($F_{(3,33)} = 0.46$, $p = 0.71$), thus further suggesting that neural reactivation was enhanced by the associated reward value, independently of the type of game. Altogether, these results provide direct evidence for the privileged reactivation, during N3 sleep, of a brain state associated with playing a game that led to a positive outcome.

To further investigate whether game-related reactivations coincided with periods of increased low delta activity, as expected from current models of memory reactivation during sleep[1], we extracted the power values for the relevant frequency band (1–2 Hz) from the EEG data of the whole sleep session, and conducted correlations between these values and the time courses of likelihood for the 5 different states detected in the fMRI data from each participant. The resulting correlation values entered an ANOVA with Brain state (Reward, No-Reward, pre-R, pre-NR, Rest) and Frequency band (low delta, 1–2 Hz; high delta, 2–4 Hz; theta, 4–7 Hz; alpha, 8–10 Hz; sigma, 12–14 Hz; beta, 15–25 Hz) as within-subjects factor. We observed a main effect of Brain state ($F_{(4,48)} = 8.51$, $p < 0.001$), a main effect of Frequency band ($F_{(5,60)} = 13.59$, $p < 0.001$), and an interaction between both factors ($F_{(20,240)} = 15.95$, $p < 0.001$). More specifically, a direct comparison between Reward and No-Reward states (planned comparison)

showed a significant difference ($F_{(1,12)} = 5.54$, $p = 0.03$) due to the positive correlation between low delta activity and the strength of the reactivation of the brain state associated with the successful game (Fig. 3c, Supplementary Table 4). No such difference was found when exploring correlations of game-related states with other frequency bands (Supplementary Table 4).

**Similar game-related activity during wakefulness and sleep.** We next asked whether the reactivation during N3 sleep primarily involved regions activated during task performance at wake (i.e., those used for the training of the classifier), or whether it may have also involved regions outside of these networks. We thus included the likelihood values for the task-specific brain states (face, maze) for each fMRI volume during N3 sleep as regressors in a whole-brain regression analysis, independently of the reward status. We obtained two sets of regions significantly activated during sleep when participants reactivated preferentially each game state, i.e., Face state vs. Rest state in N3, and Maze state vs. Rest state in N3. We then performed a conjunction analysis with the similar contrasts during the game session at wake, i.e., face game vs. rest at wake, and maze game vs. rest at wake. We could confirm that the occurrence of the game-related states during deep sleep activated a subset of regions involved when playing at each of these games while awake (Fig. 4, Supplementary Table 5). During N3, the states associated with both games reactivated a large part of the visual cortex, like playing the games did during wakefulness. However, the Face state significantly activated fusiform and occipital face-selective regions while the Maze state activated the para/hippocampal regions, consistent with

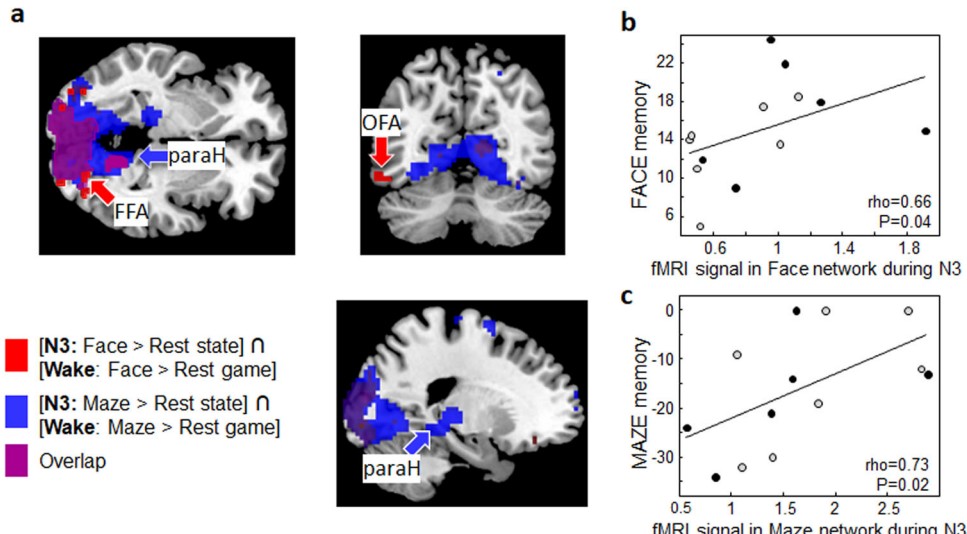

**Fig. 4 Game-specific reactivation during N3 and subsequent memory (N = 13). a** In red, brain regions more activated for Face than Rest brain state during N3 sleep, and also recruited during the execution of the task at wake (blocks of Face vs. Rest; conjunction analysis). This network included, among other regions, early visual cortices and the fusiform and occipital face regions. In blue, brain regions more activated for Maze than Rest brain state during N3 sleep, and also recruited during the maze game (blocks of Maze vs. Rest; conjunction analysis). This network included, among others, early visual cortices, the hippocampus and parahippocampal cortex. In purple, overlap between those face and maze reactivation networks during sleep. **b** Correlation between activity in the face network (red regions in **a**) during N3 and memory performance for the face game. **c** Correlation between activity of the maze network (blue regions in a) during N3 and memory for the maze game. Gray dots for participants who won the face game; black dots for participants who won the maze game. FFA fusiform face area; paraH: parahippocampus, OFA occipital face area. For memory performance measurements, see "Methods". Source data are provided as a Source Data file.

task-specific neural reactivation during sleep (see Supplementary Fig. 3).

**Task-related neural reactivation in sleep boosts consolidation.** Previous studies using cueing procedures (or "targeted memory reactivation") have suggested that task- or stimulus-specific neural reactivation early in the sleep night or during a nap enhances memory consolidation[9,20–22]. Although our participants did not spent a full night of sleep in the scanner, we tentatively sought to test for such an effect. We therefore asked our participants to come back to the lab after one full night of recovery sleep to perform memory tasks related to each of the games (Supplementary Fig. 2). We observed no overall difference in memory performance for the rewarded and non-rewarded games (ANOVA with z-scored memory performance, $F_{(1,11)} = 0.008$, $p = 0.93$). We then extracted the mean beta values from each entire reactivation network, separately for Face and Maze states, and correlated these values with subsequent memory performance for each game. We observed significant positive correlations with memory performance related to the corresponding game (Spearman correlation; face: rho = 0.66, $P = 0.04$; Fig. 4b; maze: rho = 0.73, $P = 0.02$; Fig. 4c). While correlations do not ascertain causal relationships, this pattern of results converges with cell-recordings in animals[18] and human studies using targeted memory reactivation procedures[9,20,23] to support a role for task-selective reactivations in memory consolidation processes. These findings also suggest that early post-learning NREM sleep might be critical for the consolidation of declarative memory.

**Reactivation in hippocampus and VTA during sleep.** Because during game playing, both games engaged memory regions, and because the reward status of the games was not included in the training dataset (rewarded block not included), the classifier could not distinguish the games on the basis of activity across memory and/or reward networks. Yet, based on previous animal

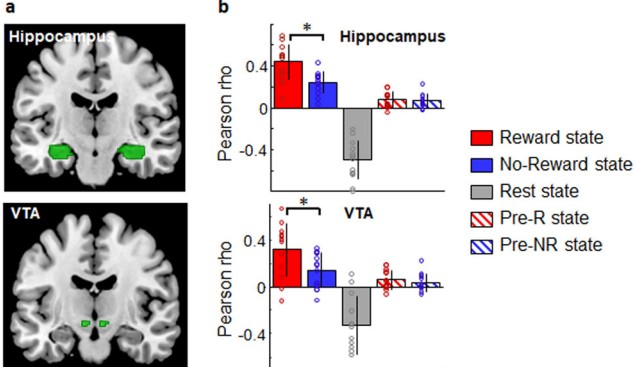

**Fig. 5 Reward-related reactivation in hippocampus and ventral tegmental area (VTA) during sleep (N = 13). a** Anatomical masks used as regions of interest (in green). **b** Mean correlation values for all participants between the time courses of the regions of interest and the different brain states. Asterisks indicate $p < 0.002$ for the hippocampus and $p < 0.02$ for the VTA for the Reward vs. No-Reward comparison, for two-sided post-hoc $t$ tests. Error bars represent SD (Standard Deviation); dots represent individual data. Repeated measures ANOVA with post-hoc test (planned comparison). Source data are provided as a Source Data file.

and human studies[12,18], we expected that the reactivation of the Reward state may be associated with an increased recruitment of memory and reward regions. We thus tested whether the hippocampus and ventral tegmental area (VTA), known to be critically involved in episodic memory and reward respectively, were preferentially engaged whenever the rewarded state was detected during sleep. To this end, we extracted the time courses of activity from two a priori anatomically-defined regions, i.e., the bilateral hippocampus (using the AAL atlas, Fig. 5a) and bilateral VTA (using a manually defined ROI on proton-density images from an independent sample of 19 participants). We then assessed the

correlation between these values and the 5 different brain states decoded during the sleep session, and performed two ANOVAs on the rho values, with the 5 states as a within-subjects factor (Fig. 5b). For the hippocampus, we observed a main effect of State ($F_{(4,48)} = 80.19$, $p < 0.001$). Post-hoc analyses revealed that the time course of hippocampal activity correlated significantly more with the Reward state than the other states (vs. No-Reward state: $F_{(1,12)} = 15.21$, $p = 0.002$; vs. Rest state: $F_{(1,12)} = 95.15$, $p < 0.001$; vs. pre-R state: $F_{(1,12)} = 80.98$, $p < 0.001$; vs. pre-NR: $F_{(1,12)} = 80.50$, $p < 0.001$). Similarly, for the VTA, we observed a main effect of State ($F_{(4,48)} = 21.0$, $p < 0.001$) and the time course of VTA activity correlated significantly more with the Reward state than the others ones (vs. No-Reward state: $F_{(1,12)} = 7.21$, $p = 0.02$; vs. Rest state: $F_{(1,12)} = 25.37$, $p < 0.001$; vs. pre-R state: $F_{(1,12)} = 22.73$, $p < 0.001$; vs. pre-NR: $F_{(1,12)} = 21.89$, $p < 0.001$).

## Discussion

Here we demonstrate the privileged reemergence of patterns of brain activity associated with a recent rewarding (compared to a non-rewarding) waking experience during sleep. Specifically, we recorded simultaneous EEG-fMRI data while participants played two games, of which only one led to a final victory, and while the same participants subsequently slept in the scanner. By first training a classifier on the awake fMRI game data and then applying it to the fMRI sleep data, we found that the brain state corresponding to the successful or rewarded game prevailed during N3 sleep, especially when slow oscillatory activity was high. We also report that the occurrence of the Reward brain state during the sleep session correlated with the activation of the hippocampus and the VTA. These findings in humans are in line with the observation in rodents that neural oscillations during slow-wave sleep coordinate the replay of recently encoded memories in the hippocampus[24], and across reward[25] and neocortical regions[26]. Post-learning increases in oscillatory activity during sleep (hippocampal sharp-wave ripples, 11–15 Hz sleep spindles, ~1 Hz SWA) are known to enhance memory retention in both humans[1,5,27–31] and animals[32,33]. Importantly, neuroimaging studies also suggested that overnight memory consolidation resulted in functional changes in brain regions specialized for the processing of task-relevant features, for example the hippocampus after a spatial navigation task[3], or the primary visual cortex after a perceptual learning task[34,35]. The present study adds experimental evidence for a close resemblance between neural activity patterns associated with specific behaviors trained during wakefulness and those spontaneously generated during sleep. In particular, our findings establish that brain states corresponding to specific waking behaviors can reemerge during N3 sleep, and further confirm that this reactivation implicates those same regions that were recruited during wakefulness (including face-selective regions for the face game and spatial navigation regions for the maze game).

Critically, here we show a privileged reprocessing of rewarded information during sleep, which could serve memory consolidation. Indeed, despite the fact that our experimental design was not optimal for testing subsequent memory (i.e., post-learning sleep was discontinued), we observed a correlation between the strength of game-related neural reactivation during early NREM sleep and subsequent memory performance. This finding may implicate a coordinated replay of hippocampal and striatal regions during sleep, as previously observed after a spatial rewarded task in rodents[18], and which could presumably contribute to a value-based consolidation of episodic memory, as recently found in humans[12]. To specifically test for the implication of the reward system during reward-biased reactivation in sleep, we used a region of interest approach and observed that

activity of the VTA increased during sleep whenever the likelihood of the Reward state was high. Thus, beyond the neural replay of recent information observed in rodents, we show that reactivation processes are regulated by the affective value associated with freshly encoded information. By combining measures of waking behavior and neural decoding during sleep, this study unveils a mechanism whereby rewards may serve as a tag to promote the reactivation and the consolidation of motivationally-relevant information during NREM sleep in humans[12–14,21]. The temporal resolution of fMRI does not allow to test for the reactivation of temporally-organized individual neurons' activity (such as during ripples), and should therefore considered as a complementary approach to single-cell recordings. The exact underlying systems and cellular mechanisms remain to be further clarified using animal models. This study also exemplifies the usefulness of neural decoding methods for testing hypotheses about the nature of spontaneous brain activity, such as fMRI data recorded during sleep, when no or little concomitant behavioral data can be collected[19].

While the current data support that neural reactivation during early NREM sleep promotes the consolidation of rewarded information, substantial evidence suggests that negative memories (and aversive conditioning) are also altered during sleep, and primarily during REM sleep when the amygdala is strongly activated[36–42]. Together these observations advocate for separate influences of rewarding and aversive events on memory reprocessing during NREM and REM sleep, as also suggested by the analysis of emotions in NREM and REM dream reports[43]. Although being unsuccessful at one of the games may have elicited negative emotions (e.g., anger, frustration) in the present study, we could not test the hypothesis of dissociated contributions of NREM and REM sleep on emotional memories with our dataset because none of our participants reached REM sleep while being scanned in the MRI. The present study therefore opens new avenues for investigating how affective values control memory playback across distinct consciousness states.

## Methods

The study protocol was approved by the Human Research Ethics Committee from the State of Geneva, Switzerland. We complied with all relevant ethical regulations for work with human subjects. All participants gave their written informed consent to take part in this study and received a financial compensation for their participation. The authors affirm that the human research participant provided informed consent for publication of the photographs in Fig. 1.

**Experimental design**. The experimental protocol involved three main visits on different days. During the first visit or *habituation phase*, participants were placed inside the MRI scanner and one structural volume was acquired (see MRI data acquisition section below). Participants were then instructed to maintain a regular sleep schedule until the second visit, which was scheduled at least 5 days later. Before leaving the lab, participants were equipped with an actimeter (Actigraph GT3X +, ActiGraph, FL, USA), which they wore on their left wrist, and asked to fill out a sleep diary including an assessment of subjective sleep quality, during at least 5 days prior to the second visit (Table S6). For the second visit or *scanning phase*, participants arrived at 6.00 pm at the laboratory and were asked to fill out the St. Mary's Hospital Sleep questionnaire assessing the quality of the previous night's sleep[44]. They ate one big sandwich, which we gave them as their evening meal, and they were then equipped with an MRI compatible EEG cap, plus EOG, EMG, and ECG (see EEG data acquisition section below). At 8.15 pm, participants underwent a short training of the tasks and were then installed in the scanner. Vigilance, attention, and anxiety states were assessed using 10-points visual analog scales (VAS; Karolinska Sleepiness Scale for vigilance[45], and scales for attention and anxiety). Next, participants were scanned while they performed the face and maze games (game session; see Tasks section below). Once the games were over, participants were taken out of the scanner and allowed to relax, walk in the lab, and eat a small snack (one piece of cake). At 10.00 pm, they were placed back inside the scanner for the sleep session. Participants again reported their levels of vigilance, attention, and anxiety. Next, they were given a 4-button MRI compatible box (HH-1 × 4-CR, Current Designs Inc., USA), which they could use to communicate with the experimenters, by pressing different buttons to tell whenever they felt too cold or too hot or uncomfortable, or whether they wanted to go out of the scanner. They

were then instructed to try to fall asleep. The sleep session lasted between 51 min and 2 h 40 min (mean: 1 h 43 min). All participants performed this session in one block except two participants, who needed a small break in the middle of the sleep session to readjust their position in the scanner. After the sleep session, we removed the EEG cap and all other electrodes, and participants took a shower. Finally, participants answered a series of questions presented as VAS about their thoughts and feelings during the game and sleep sessions (sleepiness, anxiety, physical comfort, game-related thoughts). We did not observe any correlation between the scores on any of these subjective scales and reactivation of brain states during N3 (all $p > 0.05$). Before leaving the laboratory, all participants were reminded to wear the actimeter and to fill out the sleep diary until the third visit, which was scheduled two days later (thus allowing for one full recovery night). For the third visit or *Memory test phase*, all participants came at 12.00 am to perform tests assessing their memory for elements of the face and maze games.

**Population**. Twenty-six right-handed healthy participants (18 women, 8 men, mean age ± SD: 22.0 ± 2.3 years) participated in this study. A semi-structured interview established the absence of neurological, psychiatric, or sleep disorders. All participants were non-smokers, moderate caffeine consumers, and did not take any medication. They were not depressed as assessed by the Beck Depression Inventory[46] (mean ± SD: 1.7 ± 2.0), and had low anxiety levels as assessed by the STAI-T[47] (31.8 ± 5.8). None of the participants suffered from excessive daytime sleepiness as assessed by the Epworth Sleepiness Scale[48] (5.6 ± 3.0) or sleep disturbances as determined by the Pittsburgh Sleep Quality Index Questionnaire[49] (3.1 ± 2.2). Sensitivity to Punishment and Sensitivity to Reward Questionnaire[50] established that none of the participants had extreme sensitivity to reward (37.0 ± 7.7) or punishment (32.5 ± 5.4), nor did they suffer from excessive impulsivity as assessed by the UPPS Impulsive Behavior Scale[51,52] (90.1 ± 9.6). We also made sure that none of the participants was a regular player of video games.

The data from 8 participants were removed from further analysis because of technical problems during scanning ($N = 4$), because the participants did not win at any game ($N = 2$), or because they did not sleep during the sleep session ($N = 2$). The fMRI data from 18 participants were thus used for the analysis of the game session and classifier training (12 women; mean age ± SD: 22.1 ± 2.4). Then, the trained classifier was applied to the fMRI data from the sleep session. Because we expected neural reactivation to predominate during periods of sleep with high amounts of slow oscillations, we analyzed the results from those 13 participants who had sustained N3 in the scanner (9 women; mean age ± SD: 22.0 ± 2.5). Participants who won the face and maze games (when considering the 18 and 13 participants) did not differ regarding any demographic parameter (age, depression, anxiety, sleepiness, sleep quality, sensitivity to reward/punishment, and impulsivity), and subjective VAS (vigilance, attention, and anxiety) during the scanning phase ($t$ tests, all $p > 0.05$). Moreover, they also did not differ regarding objective and subjective sleep variables (all $p > 0.05$). All statistics performed in this study are two-sided.

**Tasks**. During the game session, each participant played a face game and a maze game. The games were designed to recruit distinct and well-characterized brain networks specialized in the processing of face information (face game) and in spatial navigation (maze game). To investigate the effects of reward on neural reactivation, we manipulated the games so that, at the very end of the session, each participant would randomly win either the face or the maze game. This randomization ensured that any potential reactivation during sleep would be primarily explained by one of the games being successful (i.e., reward status), and not by some unforeseen resemblance between sleep-related activation patterns and those associated with one of the games during wakefulness. Both games have been extensively piloted so that we could manipulate which of the two games one participant would win at very end of the scanning session. Critical for the purpose of the present experiment, participants had to believe that they won one of the games thanks to their own ability, without suspecting that the games had been rigged (which actually was the case). According to a short debriefing performed after the sleep session, participants were more satisfied by their performance on the rewarded (or successful) game (6.0 ± 1.4 on a 10 point scale from 0: very unsatisfied to 10: very satisfied) than on the non-rewarded game (3.5 ± 1.4 on the same scale; $t_{(1,13)} = 5.98$, $p < 0.001$). Both games were programmed using Cogent (Laboratory of Neurobiology, University College London, UK), a Matlab-based toolbox (The MathWorks Inc, Natick, Massachusetts, US). The maze game also integrated portions of the commercial game Duke Nukem 3D (3D Realms Entertainment, Inc. Garland, Texas, US), with a homemade map. The visual rendering of Duke Nukem 3D was improved by using a high-quality texture pack (High Resolution Pack, http://hrp.duke4.net/), and the Eduke32 source port (http://www.eduke32.com/).

**Face game**. In the face game, participants had to find one target face based on clues given, one by one, by a female presenter, much like the famous "Guess Who"? game (Fig. 1a). During this task, participants were instructed to explore the 18 different faces hidden on the top half of the screen using an MRI-compatible trackball (HH-TRK-1, Current Designs Inc., USA) to move a "torch" (i.e., a circular beam of light of the diameter of one face). The faces were generated using the FaceGen software (Singular Inversions Inc, Toronto, Canada). They were clearly

distinguishable, including men and women, of various ages and ethnic origins, smiling or not, and some exhibiting specific characteristics like a tattoo or a scar. Each clue was shown as a written sentence in a cartoon bubble, as if uttered by the female presenter whose face was displayed on the lower left corner of the screen (Fig. 1a). Each clue described one feature of the target face such as "He or she is smiling" or "He or she has no visible scar". The game was split into 8 blocks (60 s each), each starting with a different clue, with both the presenter and the current clue remaining visible throughout the block.

At the beginning of each block, a new clue was presented and participants had 40 s to explore the faces with the torch before they could click on one of the faces. A green progression bar was shown and turned orange when participants could select one face. Participants had 15 s to do so before a feedback message was shown during 5 s (i.e., "This is not the correct face" or, for the last block if they won, "Congratulations, this is the correct face!"). As mentioned above, the game was rigged. After a series of poorly informative clues, the two last clues were more likely to guide participants towards the correct face, which was the presenter's face ("She is a rather talkative woman", "She stands away"). Note that the task instructions were consistent with this possibility, i.e., the preceding clues could all apply to the presenter's face and there was no mention about the correct face being one of the 18 faces. For those participants who first won the maze game, the two last clues of the face game did not point to one single correct answer ("He or She has brown eyes", "He or She has thin lips"), which prevented these participants from winning on the face game.

**Maze game**. In the maze game, participants had to find the exit of a maze, which was indicated by an "Exit" sign. Participants used the MRI-compatible trackball to navigate the virtual maze during blocks of 60 s. On each block, participants were placed at the same starting point and one additional arrow guiding them towards the exit was made visible in the maze (Fig. 1a, Fig. S2B). However none of the participants would reach the exit until block number 7. Like the face game, and unknown to the participants, the maze game was rigged. Whenever participants would first win the face game on the last blocks, the exit of the maze was blocked. The exit was difficult to find, not only because the maze was relatively large considering the allotted time, but also because an optical illusion masked the hallway leading to the exit, which would remain unexplored unless an arrow would incite participants to enter the hallway despite the illusion. On each block, participants played during 55 s and then a feedback message was shown during 5 s (i.e., "You did not reach the exit" or for the last block, if they indeed reached the exit, "Congratulations, you found the exit!").

**Timing and delivery of the tasks**. Before the main scanning session, participants underwent a brief training on both tasks: a short version of the face game with only 6 faces and 3 clues (all distinct from those used in the main game), and a short version of the maze game with a simpler map and without any arrow. There were 3 blocks for each game.

During the main scanning experiment, participants performed 8 blocks of face game and 8 blocks of maze game organized in a pseudo-random order (i.e., no more than 2 repetitions of the same game in an immediate succession). Each game block (60 s) was followed by a resting state period of 90 s, during which participants were instructed to close their eyes and let their thoughts wander. A short sound was played at the end of the rest period, indicating that participants could open their eyes and get ready for the next game block. Before each game, a screen indicated during 3 s which game was about to start (also called "pre-game" period). The whole game session lasted 40 min, divided into two fMRI runs of 20 min each.

**Delayed memory tests**. We assessed memory for elements from both games two days after the scanning phase, i.e., after one full recovery night. Memory for the face game was tested by asking participants to place each individual face at their original location on the screen. Participants were presented with a grid of 18 empty rectangles corresponding to the locations of the faces during the game. Each face was then shown at the bottom of the screen in a random order. Participants had to drag each face to the empty rectangle that they thought was its correct location. Three points were attributed for the correct location (out of 18 possible locations), 1 point was given for a correct column (out of 6 possible columns), and 0.5 point for a correct row (out of 3 possible rows; Fig. S2A). Memory for the maze game was assessed by placing participants at one specific location in the maze and asking them to find the starting point that was used during the game session as rapidly as possible. Performance was measured as the shortest map distance from the participant's current location after 30 s to the starting (here goal) point (Fig. S2B). We computed z-scores from the face and maze memory tests to be able to compare performance on both memory tasks within the same ANOVA (distances for the maze game were inversed; so that larger z-score indexed better performance). A repeated-measures ANOVA on these values with Memory task (face, maze) as within-subjects factor and Won game (participants who won at the face game and those who won at the maze game) as between-subjects factor showed no significant effect of Memory task ($F_{(1,11)} = 0.007$, $p = 0.93$), Won game ($F_{(1,11)} = 0.21$, $p = 0.65$), or interaction ($F_{(1,11)} = 1.30$, $p = 0.28$).

To test for the impact of the strength of game-related reactivation during sleep on memory consolidation processes, we performed a correlation between the level

of reactivation during N3 in task-relevant brain regions and memory performance for the face and maze games across all 13 participants (Fig. 4b). To do so, we first used the output of the classifier (organized as Face, Maze, Rest, Pre-face, Pre-maze states) as regressors in an SPM regression analysis. Next, using a conjunction procedure, we identified those voxels activated both as a function of the Face state (Face > Rest state) during N3 and when performing the face game during wakefulness, and extracted the mean fMRI activity across these voxels during the sleep session. For the maze game, we applied the same procedure, and thus identified reactivated voxels and obtained their mean activity over the course of the sleep session. We then performed Spearman correlations to test for a link between task-specific neural reactivation during sleep and delayed memory performance for each game.

## EEG

*EEG data acquisition.* EEG was continuously recorded in the scanner during the fMRI acquisition. The EEG setup included a 64-channels MRI-compatible EEG cap, two pairs of ECG, horizontal and vertical EOG, and one pair of chin EMG (BrainAmp MR plus; Brain Products GmbH, Gilching, Germany). The EEG signal was referenced online to FCz. Electrode-skin impedance was kept under 10 kOhm, in addition to the 5-kOhm built-in electrode resistance. The EEG signal was digitized at a 5000 Hz sampling rate with 500 nV resolution. Data were analog-filtered by a band-limiting low-pass filter at 250 Hz (30 dB per octave) and a high-pass filter with a 10 s time constant corresponding to a high-pass frequency of 0.0159 Hz.

*EEG data analysis.* Gradient artifacts were removed offline using a sliding average of 21 averages with the BrainVision Analyzer software (Brain Products GmbH, Gilching, Germany). Then, the signal was down-sampled to 500 Hz and low-pass filtered with a finite-impulse response filter with a bandwidth of 70 Hz. The ballistocardiogram (BCG) artifact was removed by using a sliding average procedure with 9 averages, followed by independent component analysis (ICA) to remove residual BCG along with oculo-motor components.

Two experts sleep scorers performed sleep staging on the artifact-free data from the sleep session according to AASM criteria[53] on 20 s windows of recording. Thirteen participants reached sustained N3 sleep stage. No REM sleep was obtained in the MRI.

We also conducted a power spectrum analysis of the EEG from the sleep session using the Welch periodogram method applied to successive 4-s epochs of recording from Cz, overlapping by 2 s. Data points with arousals or movement artifacts were removed before the analysis. The logarithm power spectrum was extracted. This procedure was performed for the main frequency band of interest: low delta (1-2 Hz); but also for other frequency bands for exploration purposes including: high delta: 2–4 Hz, theta: 4–7 Hz, alpha: 8–10 Hz, sigma: 12–14 Hz, beta: 15–25 Hz. We obtained 6 times series for the different frequency bands for the sleep session, which we then resampled (at the TR of the fMRI acquisition, see below) to perform correlation analyses with the brain states identified by the pattern classification method (see Pattern classification section below).

## MRI

*MRI data acquisition.* MRI data were acquired on a 3 Tesla whole body MR scanner (Tim Trio, Siemens, Erlangen, Germany) using a 12-channels head coil. Functional images were acquired with a gradient-echo EPI sequence (repetition time [TR]/ echo time [TE]/flip angle = 2100 ms/30 ms/80°) and parallel imaging (GRAPPA; acceleration factor = 2). Each functional image comprised 32 axial slices (thickness = 3.2 mm without gap, FOV = 235 × 235 mm, matrix size = 128 × 84, voxel size: 3.2 × 3.2 × 3.84 mm,) oriented parallel to the inferior edge of the occipital and temporal lobes. The two runs of the game session comprised 615 scans and 603 scans, respectively, and the run of sleep session, for the data used in the analyses reported in the main text, comprised on average 2789 scans (between 1459 and 3589 scans). The structural image was acquired with a T1-weighted 3D sequence (MPRAGE, RT/inversion time [TI]/TE/flip angle = 1900 ms/900 ms/2.32 ms/ 9°, FOV = 230 × 230 x 173 mm3, matrix size = 256 × 246 x 192 voxels, voxel size = 0.9 mm isotropic). Visual stimuli were presented on a back projection screen inside the scanner bore using an LCD projector (CP-SX1350, Hitachi, Japan), which the participant could comfortably see through a mirror mounted on the head coil. Participants responses were recorded via an MRI-compatible trackball (HH-TRK-1, Current Designs Inc., USA) during the game session and via an MRI-compatible response button box (HH-1 × 4-CR, Current Designs Inc., USA) during the sleep session.

*MRI data analysis.* Functional volumes were analyzed by using Statistical Parametric Mapping 8 (SPM8; www.fil.ion.ucl.ac.uk/spm/software/spm8) implemented in Matlab (The MathWorks Inc, Natick, Massachusetts, USA). Functional MRI data were corrected for head motion, slice timing and were spatially normalized to an echo planar imaging template conforming to the Montreal Neurological Institute (MNI) template (voxel size, 3 × 3 × 3 mm). The data were then spatially smoothed with a Gaussian kernel of 8 mm full width at half maximum (FWHM). Analysis of fMRI data was performed using a General Linear Model (GLM) approach conducted in two subsequent steps, accounting for intra-individual (fixed

effects) and inter-individual (random effects) variance. For each participant, brain responses at every voxel were fitted with a GLM, and main contrasts of interest were computed. The resulting individual maps of t-statistics were then used in second-level random-effects analyses. We used one-sample t tests to identify regions of interest for the decoding approach. Specifically, we selected peaks of activation from clusters of >50 voxels ($p < 0.001$ uncorrected) and as well as peaks in smaller clusters from well-documented task-related regions (i.e., FFA and OFA for the face game). Statistical inferences were corrected for multiple comparisons according to the Gaussian random field theory using small volumes created from anatomical regions of the AAL atlas ($p < 0.05$ corrected)[54].

For the game session, 13 conditions were modeled: "pre-face" corresponded to the 3 s period before the face game, "face-part1" when participants explored the faces (40 s), "face-part2" when participant could select one face (15 s), the rest period following the face game was divided in 3 equivalent parts (30 s each: "face-rest1", "face-rest2", "face-rest3"), "pre-maze" corresponded to the 3 s period before the maze game, 'maze' when participants explored the maze (60 s), the rest period following the maze game was also split into 3 periods of 30 s ("maze-rest1", "maze-rest2", "maze-rest3"), "WIN-block" was the last block of the won game (face or maze), and "WIN-rest" was the rest following the WIN-block. All these conditions were entered in the design matrix as separate regressors (block design) convolved with the canonical hemodynamic response function (HRF). Movement parameters estimated during realignment were added as regressors of no interest in the first-level analyses. We then used t tests across the 18 participants at wake to define regions of interest (ROIs) for the pattern classification during sleep (see Pattern classification section below).

To better characterize the brain networks corresponding to the different brain states for each game during the sleep session, we used the time course of the game-related brain states identified by the pattern classification (Face, Maze, Rest) as regressors in an SPM design matrix on those 13 participants who reached stage N3 sleep. The states Pre-face and Pre-maze were not entered in the design due to their low occurrence during the sleep session. The goal of this whole-brain analysis was to reveal regions whose activity correlated with the Face or Maze brain states (compared to Rest state) during N3, within and possibly outside of the ROIs used for the training of the classifier on the wake data. As such, this complementary analysis offered an additional test for region-specific reactivations during sleep (Fig. 4a, Table S5). To identify brain regions significantly activated both during one specific game while awake and when the likelihood of occurrence of the brain state associated with that game was high during subsequent sleep, we performed a conjunction analysis combining the contrast Face vs. Rest states during N3 (or Maze vs. Rest state during N3) and the contrast face game vs. rest during wake (or maze game vs. rest during wake).

Finally, we asked whether regions involved in the processing of declarative memory (i.e. hippocampus) and/or in reward-related dopaminergic modulation (i.e., ventral tegmental area, VTA) may show activity changes whenever the Reward (as compared to the No-Reward) state was detected during sleep. We created one anatomical mask for the bilateral hippocampus using the AAL atlas and another anatomical mask for the bilateral VTA that we delineated on proton-density images from an independent sample of 19 young males, healthy participants (mean ± SD: 22.05 ± 2.78 years old). From each of these masks, we extracted the time course of activity during the sleep session, averaged across all voxels within each mask, and computed correlations with time courses of the different states used in the classifier (Reward, No-Reward, Rest, pre-R, pre-NR) during the sleep session as well. Next, we performed separate ANOVAs for the hippocampus and for the VTA on the obtained Pearson rho values with State as within-subjects factor. When the main effect of State was significant, post-hoc planned comparison tested for differences between specific conditions. All statistics were performed two-sided.

## Pattern classification

We used a pattern recognition approach to test whether task-specific patterns of neural activity recorded during the game session could be detected in the data recorded during the sleep session. This approach involved 4 main steps for the decoding of distinct brain states during the sleep session (Fig. 1b): (1) definition of the states and ROIs based on the fMRI data from the game session, (2) training of a classifier on the fMRI data of the game session at wake extracted from the ROIs, (3) validation of the classifier using a leave-one-out procedure, (4) application of the classifier to the Sleep fMRI data, so as to get the likelihood for each brain state to be present in each brain volume acquired during the sleep session. Then, the mean likelihood of each brain state was computed for each scored period of wakefulness and sleep stages (N1, N2, N3), and for each participant. We could thus test whether the brain state corresponding to the successful game (Reward, compared to No-Reward, Rest, pre-R, or Pre-NR) had any advantage for reactivation during sleep, in particular during deeper sleep stages (see main text). We also used the time course of the likelihood for each game-related state to occur in the fMRI Sleep data in a whole-brain analysis using SPM (see previous section).

We first defined 5 relevant states for the classification of the Game data (Fig. S1): Face state (8 blocks of face game playing, 60 s; corresponding to the duration of face-part1 + face-part2 defined in the previous section), Maze state (8 blocks of maze game playing, 60 s), Rest state for the periods following each game (16 blocks of 90 s). We also added a Pre-face state (8 blocks of 3 s preceding each face game) and a Pre-maze state (8 blocks 3 s preceding each maze game), as states

of no interest, but which may yield spurious classification results if not explicitly defined. We then defined 58 ROIs based on the SPM contrasts on the Game data: face > maze, maze > face, face-rest1 < face-rest3, maze-rest1 < maze-rest3 (Table S1). The two former contrasts identified regions selectively activated for each game; the two latter contrasts were used to reveal regions whose activity was suppressed immediately following each of the games relative to later periods of rest and were also selected for the analysis of post-task resting periods not reported here. We based this approach on our previous work in which we demonstrated that those regions whose activity was significantly suppressed immediately after the presentation of video clips (first 30 s relative to subsequent 30 s temporal bins) largely overlapped with those activated during video watching[55]. Here, and as expected, these contrasts (for face and for maze) mostly engaged task-selective regions (see Supplementary Table 1). To best delineate the network of regions involved in each of the games, we thus combined all regions whose activity increased during game playing and/or decreased for the first compared to the last Rest temporal bins. We extracted the time-course of fMRI activity from spheres (radius of 5 mm) centered on the peak of activation for each ROI, and let the classifier assign one of the five labels (i.e., brain states) to each time-point for the data from all participants who performed the game session ($N = 18$). Please note that, as for all main contrasts (see above), the extracted time-courses of fMRI activity did not include the data from the block during which participants won one of the games (and after which the game ended). We ran the validation of the classifier using a leave-one-out procedure. As shown on Fig. S1, classification accuracy was high for the Face, Maze, and Rest states (all > 0.76), but low for the Pre-face and Pre-maze states. The classifier was then applied to the Sleep data of the participants who reached N3 sleep ($N = 13$) (Fig. 3a). To test for the selective reactivation of the successful game, we conducted an ANOVA on the mean likelihood of occurrence of each brain state during distinct sleep stages, with 5 Brain states defined now as a function of which game has been won by each participant (Reward, No-Reward, Rest, pre-R, pre-NR) and 4 Sleep stages (Wake, N1, N2, N3) as within-subjects factors (Table S3). We also performed post-hoc analyses for significant interactions.

Second, because current models of memory replay during sleep suggest a critical role for slow waves (delta activity) in orchestrating coordinated reactivations[1,28], we computed correlations analyses (Spearman correlation, with Fisher z-transform) between the likelihood of occurrence of each brain state (Reward, No-Reward, pre-R, pre-NR, Rest) and the power spectrum of low delta (1–2 Hz) across time. For exploratory purposes, we also investigated other frequency bands (high delta, 2–4 Hz; theta, 4–7 Hz; alpha, 8–10 Hz; sigma, 12–14 Hz; beta, 15–25 Hz; Table S4). We subdivided the delta band in two subcomponents (low and fast delta) to take into account possible functional distinctions between both subcomponents[56]. The rho correlation values for each participant were analyzed using a repeated-measures ANOVA with Brain state and Frequency bands (high delta, theta, alpha, sigma, beta) as within-subjects factors. Post-hoc analyses were used to decipher significant interactions using planned comparisons.

**Classification method**. We used a classifier based on Conditional Random Fields (CRFs), whose predictions can account for temporal context[57,58], such as the different event episodes in our fMRI task paradigm (e.g., face and maze blocks)[59]. The fMRI time courses of the $1,..., N = 58$ ROIs were embedded into a feature vector $x_t = [f_1, f_2, ..., f_N]$ for all $T$ timepoints $t$. We used the following $S = 5$ possible labels $y_t$ of the GLM conditions (i.e., Rest, Pre-maze, Pre-face, Maze, Face) to perform supervised learning. In particular, we deployed the conventional conditional random field (CRF) that models all labels of the observation data with one exponential distribution[60]. Based on an underlying graphical model, one can define the following product of potential functions for a label sequence $y$ and an observation sequence $x$:

$$\exp\left(\sum_{i,j}^{S} \lambda_{i,j} t_{i,j}(y_{t-1}, y_t, \boldsymbol{x}, t) + \sum_{k}^{S} \lambda_{i,j} t_{i,j}(y_{t-1}, y_t, \boldsymbol{x}, t)\right) \quad (1)$$

where $t_{i,j}$ is a transition feature function and $s_k$ a state feature function. The set of weights $\boldsymbol{\theta} = [\lambda_{1,1}, \cdots, \lambda_{S,S}, \mu_1, \cdots, \mu_S]$ needs to be estimated from the training data. By stacking all transition and state functions as $\boldsymbol{\varphi} = \left[t_{1,1}, \cdots, t_{S,S}, s_1, \cdots, s_S\right]$, we can then rewrite the above equation using the inner product as $\exp(\langle\boldsymbol{\theta}, \boldsymbol{\varphi}(\boldsymbol{x}, \boldsymbol{y})\rangle)$. Then, the probability of $y$ given an observation $x$ is obtained as

$$p_{\boldsymbol{\theta}}(\boldsymbol{y}|\boldsymbol{x}) = \frac{\exp(\langle\boldsymbol{\theta}, \boldsymbol{\varphi}(\boldsymbol{x}, \boldsymbol{y})\rangle)}{Z_{\boldsymbol{\theta}}(\boldsymbol{x})} \quad (2)$$

where $Z_{\boldsymbol{\theta}}(\boldsymbol{x}) = \sum_{\boldsymbol{y}} \exp(\langle\boldsymbol{\theta}, \boldsymbol{\varphi}(\boldsymbol{x}, \boldsymbol{y})\rangle)$ is the normalization factor.

The log-likelihood that is maximized to fit the CRF model is defined as

$$L(\boldsymbol{\theta}, \text{data}) = \sum_k (\langle\boldsymbol{\theta}, \boldsymbol{\varphi}(\boldsymbol{x}^{(k)}, \boldsymbol{y}^{(k)})\rangle - \log(Z_{\boldsymbol{\theta}}(\boldsymbol{x}^{(k)}))) \quad (3)$$

where $\boldsymbol{x}^{(k)}$ is the $k$-th observation sequence and $\boldsymbol{y}^{(k)}$ the $k$-th label sequence of training data. Fitting is done using iterative techniques described in[61]. The fitted model is then matched to an observation sequence of the test data using the Viterbi algorithm[61].

We fitted the CRF model to the task data and performed leave-one-session-out cross-validation (across subjects). Since we aimed at ultimately applying the

CRF model to the sleep data for which the temporal sequence of the paradigm is not necessarily meaningful, we kept the transition feature functions constant and uniform. The confusion matrix for the cross-validation on the data of the game session is shown in Supplementary Fig. 1. All states are well decoded, except the Pre-maze and Pre-face ones that are confounded with the Rest condition. Next, we applied the same CRF model to the data from the sleep session. Finally, although very unlikely given the randomization of the game type across reward conditions, we wanted to ensure that higher reactivation of the rewarded task during sleep did not trivially relate to better performance of the classifier, as assessed during wakefulness. We thus used the mean probability or likelihood time courses for the 5 states, now according to which game was won (Reward, No-Reward, Rest, pre-R, pre-NR) and tested for correlations between decoding accuracy during the game session and the strength of state reactivation during sleep. No significant correlation was found (Reward: $R = -0.24$, $p = 0.43$; no-Reward: $R = 0.48$, $p = 0.09$; Rest: $R = -0.31$, $p = 0.31$; pre-R: $R = 0.02$, $p = 0.95$; pre-NR: $R = 0.42$, $p = 0.15$). These results suggest that the probability of reactivation during sleep did not relate to how accurately the classifier classified the game- or rest-related states during the training (game) session.

**Reporting summary**. Further information on research design is available in the Nature Research Reporting Summary linked to this article.

## Data availability
The raw data sets generated and analyzed during the current study are available in the openneuro.org repository https://openneuro.org/datasets/ds003574/versions/1.0.2 [62]. Source data are provided with this paper.

## Code availability
MRI preprocessing and MRI analysis were performed using SPM scripts that are publically available on the SPM website (https://www.fil.ion.ucl.ac.uk/spm/). The codes related to the conditional random fields are available as public open source codes (CRF++: Yet Another CRF toolkit. https://taku910.github.io/crfpp/). All codes used to run the analysis are also available from the authors upon request.

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

## Acknowledgements

This research was supported by the National Center of Competence in Research (NCCR) Affective Sciences financed by the Swiss National Science Foundation (grant number: 51NF40-104897) and hosted by the University of Geneva, and the Swiss National Science Foundation (320030-159862 and 320030-135653), and the Mercier Foundation. Hee-Deok Yang work was supported by the National Research Foundation of Korea (NRF) grant funded by the Korea government (MSIT) (No. NRF-2017R1A2B4005305). We also thank Ben Meuleman for statistical advice.

## Author contributions

V.S., S.S. designed the experiment. V.S., M.v.S., M.C., and A.R. conducted the experiment and acquired the data. V.S., M.v.S., M.C., A.R., S.P., H.D.Y., D.V.D.V., and S.S. analyzed the data. V.S., H.D.Y., D.V.D.V., and S.S. wrote the manuscript.

## Competing interests

The authors declare no competing interests.
