## [Peer Review File · Nature Communications]

Reviewers' comments:

Reviewer #1 (Remarks to the Author):

The authors examined the role of reward in biasing spontaneous reactivations signals during sleep in healthy humans. They trained a classifier on brain signals acquired by fMRI recordings during performance of two different complex games, with one of them being rewarded. Subsequently, they applied the trained classifier on fMRI data acquired during sleep. They report preferential activation of the rewarded as compared to the non-rewarded game during sleep.

The general topic of the paper is highly relevant and very interesting. The methodological approach is very sophisticated and goes beyond the state-of-the art. Unfortunately, the number of examined subjects is quite low, which might question the robustness of the reported results. The authors need to increase their sample size to increase the trustworthiness of their data.

Major points:

- The number of participants in the main analysis is quite low. From the 18 participants, only 13 are included in the main analysis. In this analysis, $n = 7$ participants are in one group (winning the FACE game), whereas $n = 6$ participants are in the second group (winning the MAZE game). The authors should at least increase the number of participants to $n \geq 10$ in each group. In addition, they need to use the same number of participants in each analysis. If I understand correctly, the analysis of the learning phase included all 18 participants, whereas the analysis of the sleep phase was restricted to 13 participants.
- Please explain in more detail why the classifier was trained on 5 states, but was restricted to three states when applied on data acquired during sleep. Or did the classifier classified also 5 states during sleep, but only three were used for further processing? In the first case, the authors appear to force the classifier to decide between waking rest, MAZE reactivation and FACE reactivation. However, the most likely case is that none of the three categories occurs during sleep. Thus, the general increase in the two reactivation labels during sleep might simply occur because brain activity during sleep is not like waking rest. Please comment on and discuss this issue. Ideally, please repeat classification performance during sleep with a fourth label and report the results.
- The correlation between fMRI signal and memory reported in Figure 3 appear to be unusually high ($r = 0.9$ and $r = 0.7$). Please make sure the no non-independent analysis steps are used during the analysis (see Vul et al., 2009, *Perspect Psychol Science*). In addition, please your conclusion in this context, as correlations provide not evidence for a causal role (line 215).

Minor Points:

- Supplementary Table 1: Please provide the accuracy of the trained classifier during the Game Session for each individual subject. Do the individual differences in classifier accuracy during the Game session relate to the results during sleep? Furthermore, please add standard deviation to the average classification performance.
- Table 1: Please indicate statistical differences between the two experimental groups.
- Line 195: Please use some statistical procedure to confirm any overlap between task-related and reactivation-related brain activation. So far, the conclusion of "task-selectivity" appears not be supported by any statistical finding. In addition, the procedure of ROI selection is unclear. Which anatomical regions of the AAL atlas were taken and why? Did the authors have any a priori hypothesis on these anatomical regions? Why did they not take the results pattern during learning as one ROI? What do the authors exactly mean by small volume correction in Supplementary Table 4?
- Supplementary Information, line 217: Please explain in more detail how the main task-related activations were defined: Which thresholds were used on the probability level or cluster level to define the number of peak voxels later used in the ROI definition? Please add this information also in the MRI Studies Reporting Summary.

Reviewer #2 (Remarks to the Author):

This is an interesting and well designed study that addresses timely questions about the role of sleep in enhancing behaviorally relevant memories. In the experiment, human participants engaged in a series of computer games that ended with winning either a face game, or a maze game. The authors exploited the localization of maze vs. face related brain activity to classify fMRI activity during the game, and then used the same classifier on fMRI data collected while participants later slept. This approach revealed that during sleep there was more activity classified in the category of the game that participants won and that this was the case particularly in the N3 phase of sleep.

The paper is clearly written and the data potentially provide novel evidence bearing on mechanisms of replay during sleep and its role in memory consolidation. My main concern about the study is the sample size. The final analyses linking N3 activity (putative replay) with reward - the key findings of the paper - are performed on a very small sample of 7 and 6 participants in each condition. I recognize that this study is logistically very challenging, but I am concerned about basing conclusions on such a small sample. At the very least, it would be crucial to show individual data to determine how many of the 13 participants showed a difference between the reward and no-reward games during N3 sleep.

Adding to this concern, in several places the analyses demonstrate a significant difference in one condition (e.g. N3) but not the other phases, but without a direct statistical comparison between them (e.g. line 163/Figure 3A, re significant interaction only in N3; line 215 re memory effects with the corresponding game, but not the other game).

Figure 3CD: it would be helpful to visualize which of these 13 individual data points corresponds to Ss who won the maze vs. the face game.

Reviewer #3 (Remarks to the Author):

General comment:

Sleep has a well-documented role on memory consolidation. One of the proposed mechanisms is that memories are replayed during sleep, such replay leading to the strengthening of mnemonic traces. While replay has been widely observed in rodents, results are scarce in Humans. In addition, it is still unclear how replay is organized. Here, Sterpenich and colleagues argue that rewarded representations are more likely reactivated during sleep. According to the authors, this could represent a central mechanism allowing the selection and consolidation of the most valuable memories.

This study thus represents a potentially very important step forward. By demonstrating that memory reactivation is not blind but could be biased by memories' own relevance, the authors would provide a crucial addition to the current model of active consolidation (Diekelmann & Born, TICS 2010). However, I have serious concerns about the study's methodology and authors' interpretation of their results. Notably, I have the feeling that the study is underpowered and requires additional data.

Major comments:

1. Results' interpretation

1.1. The authors try to link the reactivation they observe with the hippocampal replays. Nonetheless, they do not point out at the major differences between these two phenomena. Hippocampal replay are short (few hundred milliseconds) and localized in hippocampal regions whereas, here, they observed the reactivation of patterns of brain activity across several cortical regions and at much larger time-scales (since they used the BOLD signal). It is quite hard to link these two phenomena. Indeed, the condensed nature of hippocampal replays allow them to be embedded in sleep oscillations (ripples, spindles and slow-waves) which is deemed crucial for the consolidation of the associated memory. If the representation of the games were here replayed during sleep with positive consequences on subsequent memory, as the authors claim, this must be done through very different mechanisms than as for hippocampal replays. This should be discussed in the manuscript.

1.2. The authors claim that they 'uncover[ed] a neural mechanism whereby the most valuable elements of our lives become engraved on our memory while we sleep' (l. 35). This study actually falls short at providing such mechanism. The authors do show that representations associated with rewards are more likely to be reactivated in sleep, but they don't show or propose any neural mechanisms explaining how this is done. It would have been interesting to see how the re-activations occurring during N3 relate to the reward-circuits and what triggers the reactivation of the winning game.

2. Methodology

2.1. I find the authors' choices to train and validate their classifier surprising. They indeed trained the classifier on 5 classes that partly overlapped. Indeed, the Pre-FACE and Pre-MAZE states seem quite close to the REST state. In addition, the Pre-FACE and Pre-MAZE states comprised 8*3s of data whereas the FACE and MAZE states comprised 8*60s of data and the REST state 16*90s! It seems that this choice of unbalanced and overlapping classes can only add noise to the fitting of the classifier. Accordingly, classifier's validation (Fig. S1) shows that the Pre-FACE and Pre-MAZE classes were not correctly decoded. I would advise the authors to fit the classifier on three classes only (FACE, MAZE and REST). As a side note, the authors should quantify the performance of the classifier for each class and overall using F1-scores or ROC-curves.

2.2. The authors used a between-subjects design in which they crucially tested the significance of a triple interaction between two within-subjects factors (Brain-state and Sleep-stage) and a between-subject factor (Group). Surprisingly, they only have 6 and 7 participants in the two Groups. The study seems thus quite underpowered. The authors should check that (i) their data meet the requirements of the repeated measure ANOVA despite the small sample size, (ii) they have sufficient power to check for the triple interactions. If this is not the case, the authors will have to modify their analyses and/or acquire new data after estimating the sample-size needed.

2.3. Given the small sample-size in the two groups (n=6 and 7), the authors should avoid the use of bar-plots (e.g. in Figure 3), especially since they did not define the nature of the error-bars used. The authors should plot the individual data. The authors should also focus on one sample (i.e. the subjects with enough N3) for clarity and should, in any case, always mention the sample-size for all figures and statistical tests.

2.4. The authors investigated the relationship between game-related re-activations and delta activity. However, from the different regions showing differential activations between the FACE (or MAZE) state and REST (see ST4), they report only the results for the regions leading to significant results (OFA and hippocampus, Fig. 3 C-D), which is quite close to cherry-picking the results. Since they tested many more regions and frequency ranges (l. 193-194), they should correct their results for multiple comparisons. I am not sure the results would hold with such constrains.

2.5. In the following analysis (l. 195-203), the authors seek to check which regions were involved during the N3 re-activations: 'We thus included the likelihood of each brain state for each fMRI volume during N3 sleep as a regressor in a whole-brain regression analysis' (l. 197). They observed significant activations within the task-related ROI. This is quite trivial and akin to double-dipping. Indeed, the classifier was trained on specific ROIs. The likelihood for the MAZE pattern, for example, would therefore increase when the MAZE ROI get activated. Thus, finding activations during N3 that match the initial ROI could just reflect how the classifier was trained. In my opinion, it does not allow to conclude on any 'task-selectivity' (l. 202) or region-specificity (l. 196).

2.6. In the main text, most of the statistical tests are not properly described (test name, sample-size, effect-size...), which makes it impossible to assess the reliability of the statistical analyses.

Minor comments:

1. I find the definition of the REST contrast and ROI puzzling: 'the two latter contrasts were used to reveal regions whose activity was suppressed immediately following each of the games relative to later periods of rest' (SI: l. 261). Why focusing on regions suppressed after each game? I would expect a REST-state to cover mostly regions within the Default-Mode Network. How do the ROI for the REST state overlap with the DMN? This would be important to understand why the REST state

tends to disappear after sleep onset.

2. In the manuscript, the authors should draw a clear distinction between the notion of replay (as seen in rodents: a clear repetition of a sequence of firing condensed in time) and the reactivation of patterns of brain activity similar to the game-related activation as observed here.

3. Along the same line, the authors wrote that 'This finding may implicate a coordinated replay of hippocampal and striatal regions during sleep, as previously observed after a spatial rewarded task in rodents [16]' (l. 243). However, the reference cited here (Lansik et al. 2009) showed a much different form of replay than the reactivation evidence here (short replay associated to hippocampal ripples). In addition, the authors did not show here any form of coordination between hippocampal and striatal regions.

4. Would not it be preferable to exclude the last blocks from the classifier since these blocks could be contaminated by activation associated to the fact of winning?

5. Why using a repeated measure ANOVA (with strong assumptions on the equality of variances hard to meet with small sample sizes) and not non-parametric method?

6. Why was memory performance z-scored? Could the authors also provide an analysis of the raw memory performance (in particular were the participants above chance-level two days after the game session?)?

7. Why did the authors divide the delta-band in two? Why choosing a lower boundary at 1Hz and not 0.5Hz, a more classical value?

8. 'No overall difference' (l. 207) Could the authors specify the tests performed and the results obtained?

9. The authors wrote: 'The present study adds unprecedented experimental evidence for a close resemblance between neural activity patterns associated with specific behaviors trained during wakefulness and those spontaneously generated during sleep' (l. 233). The authors did not really show a 'close resemblance'. How close the reactivation is from the original activation is assessed relatively to other states. It could be quite far and still discriminate the different states. Similarly, the authors should avoid terms as 'strongly regulated' as it is difficult to assess the strength of the results with such a small sample-size.

10. 'further confirming that this reactivation implicates those same regions that were recruited during wakefulness' (l. 236). See also major comment 2.5, this cannot be concluded from the analysis performed.

11. In the last paragraph, the authors advocate for separate effects of REM and NREM sleep for the reactivation of negative or positive memories respectively. This is very speculative since (i) such contrast has not been directly tested, neither here nor in the work cited, (ii) as the authors mention, the subjects may have experienced negative emotions as well, associated to the losing game.

12. Regarding Table ST1, could the authors perform an analysis of potential differences between the two groups (FACE-win, MAZE-win)?

13. Could the authors provide the scores of participants in the different questionnaires made (SI: l. 41-48)?

14. 'participants had to believe that they won' (SI: l. 67). Was this checked? How?

15. 'EEG data analyses': what is 'a sliding average of 21 averages' (SI: l. 166)?

16. It is weird for the authors to mention that they designed a classifier with the expectation that it would fail to classify: 'classification accuracy was high for the FACE, MAZE, and REST states (all

> 0.76), but low for the pre-FACE and pre-MAZE states, as expected' (SI: l. 267-269).

17. Table ST3: were correlations for sigma band significant?

Reviewer #1

Comment #1

- The number of participants in the main analysis is quite low. From the 18 participants, only 13 are included in the main analysis. In this analysis, $n = 7$ participants are in one group (winning the FACE game), whereas $n = 6$ participants are in the second group (winning the MAZE game). The authors should at least increase the number of participants to $n \geq 10$ in each group. In addition, they need to use the same number of participants in each analysis. If I understand correctly, the analysis of the learning phase included all 18 participants, whereas the analysis of the sleep phase was restricted to 13 participants.

Response #1

We would like to apologize for having evidently and regrettably misled this and the other reviewers by presenting our data as if we may have aimed at comparing the results from two independent groups of participants. If this were indeed our aim, we would concur on the reviewers' concern about sample size. Note however that the data and methods reported in this study would actually allow looking at relatively small samples, as we discuss below. In fact, and as also stated in the initial submission, the goal of our study was to show that patterns of neural activity corresponding to one type of waking experience -- here playing one specific game -- were more likely to be spontaneously reactivated during NREM sleep if the game had been rewarded as compared to a non-rewarded game. We used a neural decoding approach to assess the likelihood of reactivation of distinct brain states specifically associated with playing the rewarded game and the non-rewarded game (and other states as well) during sleep. Because we could not exclude *a priori* that spontaneous patterns of brain activity during NREM sleep may intrinsically resemble those elicited when participants played one of the two games (i.e. face or maze), we carefully controlled for this possible confound by randomizing the reward factor across the games. This randomization allowed us to ascertain that greater reactivation of the brain state corresponding to one game during NREM sleep was attributable to the associated reward-value and not to the type of game itself. Therefore, our experimental design is a within-subjects design, including 13 participants in whom we compared a Reward (R) and No-Reward (NR) condition, and not 7 and 6 in each relevant experimental condition. To illustrate that this procedure successfully controlled for any spurious effect of the type of game, we initially presented separately the results from the participants who were assigned either game. We realize that this figure may have misled the reviewers and should have been presented as supplementary material. Therefore, we now present the main results for the whole sample of participants in Figure 3 (replicated here below). In the original manuscript, we also reported an ANOVA with two within-subjects factors (Brain State and Sleep Stage) and one between-subjects factor (Type of Game), which may have further erroneously suggested that we aimed at comparing participants as a function of the specific game that was rewarded. We now describe the main analyses without this factor. We also show that 11 out of 13 participants exhibited higher likelihood values for the brain state corresponding to the rewarded game than the state corresponding to the non-rewarded game (see Figure 3 below), with one participant showing the reverse pattern and one showing very little reactivation.

Figure 3. Classification results (N=13). (A) Mean likelihood of Reward, No-Reward, Rest, pre-Reward (pre-R) and pre-No-Reward (pre-NR) brain states for each sleep stage, showing increased reactivation of the brain state associated with the rewarded game during N3. (B) Individual data for the Reward and No-Reward state during N3 sleep. (C) Correlation between time course of likelihood and low delta power (1-2 Hz) for the Reward and No-Reward states. Error bars represent SEM; dots represent individual data points.

Specifically, a repeated-measure ANOVA on likelihood measures with Brain state (Reward, No-Reward, Rest, pre-R, pre-NR) and Sleep stage (wake, N1, N2, N3) as within-subjects factors showed main effects of Brain state ($F_{(4,48)}=137.1$, $P < 0.001$), and an interaction between Brain state and Sleep stage ($F_{(12,144)} = 14.8$, $P < 0.001$). Post-hoc comparisons first showed that the Rest state predominated over the Reward and No-Reward states during wakefulness (Rest vs. Reward: $F_{(1,12)}=263.2$, $p < 0.001$, Rest vs. No-Reward: $F_{(1,12)}=515.8$, $p < 0.001$), while it was progressively less present from light to deeper sleep stages (main effect of sleep for the Rest state: $F_{(1,12)}=371.9$, $p < 0.001$). These results suggest a similarity between resting blocks during the game session and resting while awake during the sleep session. Conversely, both game-related brain states had a relatively low likelihood to occur across all sleep stages (less than 0.20), and did not differ between Reward versus No-Reward states from wakefulness to N2 (R vs. NR for wake: $F_{(1,12)}=2.24$, $p=0.16$, N1: $F_{(1,12)}=0.18$, $p=0.67$, N2: $F_{(1,12)}=0.03$, $p=0.87$). By contrast, and confirming our main hypothesis, the Reward state differed from the No-Reward state during N3 sleep ($F_{(1,12)}=10.65$, $p=0.007$). The states corresponding to the preparation period before each game did not differ for the rewarded and non-rewarded games in any sleep stage (pre-R vs. pre-NR states for wake: $F_{(1,12)}=0.002$, $p=0.96$, N1: $F_{(1,12)}=0.37$, $P=0.56$, N2: $F_{(1,12)}=0.03$, $p=0.88$, N3: $F_{(1,12)}=1.42$, $P=0.26$). These results are now reported in Results section of the revised manuscript.

Concerning the number of participants included in the analyses: the classifier was first trained on the fMRI data acquired during the game session from all participants who performed the entire experimental protocol, i.e., winning at one game and sleeping in the scanner (18 participants). As for

the ANOVA performed on the data acquired during the sleep session, because we expected slow oscillations to favor task-related neural replay, we only included those participants who reached consolidated N3 sleep, i.e. 13 participants. Training on a large sample of wake data and then applying the classifier on a subset of relevant sleep data thus allowed to exploit as much of the available data as possible. Moreover, if anything, such a procedure should improve the generalizability of any significant result.

Comment #2

- Please explain in more detail why the classifier was trained on 5 states, but was restricted to three states when applied on data acquired during sleep. Or did the classifier classified also 5 states during sleep, but only three were used for further processing? In the first case, the authors appear to force the classifier to decide between waking rest, MAZE reactivation and FACE reactivation. However, the most likely case is that none of the three categories occurs during sleep. Thus, the general increase in the two reactivation labels during sleep might simply occur because brain activity during sleep is not like waking rest. Please comment on and discuss this issue. Ideally, please repeat classification performance during sleep with a fourth label and report the results.

Response #2

We thank the reviewer for raising this issue, which prompted us to clarify an important point. The classifier was trained on the 5 states, and all 5 states were then used during the classification of the sleep data. This process avoided forcing the classifier to choose between only 3 states (among which 2 corresponded to game-related activity). The performance of the classifier during wakefulness is plotted below (Figure S1, replicated below). The classifier's accuracy was computed on the 5 relevant states for the wake data, namely the brain states corresponding to the period preceding each game, the states for the face and maze games proper, and the states corresponding to the resting periods. Please note that, for each participant, the reward (and no-reward) status was conferred to one of the games at the very end of the games and the data corresponding to the winning blocks were not included in the training set.

We included all 5 distinct states for both the training on wake data and the subsequent classification of fMRI data during sleep. We did this precisely to avoid that the classifier may be artificially biased towards one specific state, whenever the brain state displayed by one fMRI image would strongly differ from other states. In other words, any data point that would differ from Rest state would randomly distribute across the 4 remaining states and not only across "game-related states". In the figures, we initially plotted only the 3 main states of interest (Rest, Reward, No-Reward) for comprehension and simplification. We realize that this may have confused the reader and now report the data for all 5 different states (see Figure 3 above).

On Figure S1B below, the most represented state is Rest, and its occurrence decreased with sleep depth. This pattern is consistent with Rest state during wake and light sleep in the sleep data being most similar to those data from the rest periods during the games with eyes closed (as we would expect), thus further validating the interpretability of the classifier's results. Please note also that the low accuracy of the classifier for the pre-R and pre-NR states is probably due to the short duration of both these conditions as compared to the other (3 sec for the pre-game versus 60 sec for the games and 90 sec for the rest period). We would like to stress that even if some unlikely bias may have

affected the classification of the Rest state vs. the replay states, the critical comparison in the current study was between the Reward and No-Reward brain states, which should not have been affected by any such bias (also controlled by the randomization of the game-type across reward conditions). Moreover, the whole brain regression analysis performed with the state likelihood for the sleep data further confirmed that brain regions related to playing at one of the game were indeed activated during sleep whenever the likelihood of this game state was high (see our reply to comment #3 below).

Figure S1: Task-related decoding accuracy. (A) Confusion matrix for the 5 states classified using the fMRI data of game session. Decoding accuracy resulted from a leave-one-out procedure and indicated high classification accuracy for the Face, Maze, and Rest states. (B) Individual measures of decoding accuracy for predicted vs. actual state (N=18).

In the revision, we amended the description of the procedure in the main text, as follows:

“Next, we trained a classifier to dissociate between 5 distinct brain states corresponding to playing the (to-be) rewarded and non-rewarded games (either face or maze game according to participants), and to the blocks of rest (Fig. 1B). We also included two states of no interest, pre-rewarded (pre-R) and pre-non-rewarded (pre-NR), which corresponded to the 3 seconds preceding each game; see Supplementary Information).”

We also amended the corresponding section in the Supplementary Information file as follows:

“To test for the selective reactivation of the successful game, we conducted an ANOVA on the mean likelihood of occurrence of each brain state during distinct sleep stages, with 5 Brain states defined now as a function of which game has been won by each participant (Reward, No-Reward, Rest, pre-R, pre-NR) and 4 Sleep stages (Wake, N1, N2, N3) as within-subjects factors (Table S3). We also performed post-hoc analyses for significant interactions.”

Comment #3

The correlation between fMRI signal and memory reported in Figure 3 appear to be unusually high ($r = 0.9$ and $r = 0.7$). Please make sure the no non-independent analysis steps are used during the

analysis (see Vul et al., 2009, *Perspect Psychol Science*). In addition, please your conclusion in this context, as correlations provide not evidence for a causal role (line 215).

Response #3

We modified our approach to correlating brain results to subsequent behavior to refrain from selecting a-priory regions-of-interest. Instead, to delineate brain regions showing game-related reactivation during sleep, we now retained all brain regions commonly activated for the contrast face (or maze) vs. rest during both the awake game session and N3 sleep (i.e. Face or Maze state vs. Rest state) using conjunction analyses (see Figure 4A replicated below). We could confirm that the occurrence of the states related to each game during deep sleep activated a subset of regions involved when playing at the corresponding game while awake (Figure 4 in red and blue, see also revised Supplementary Table 5). During N3, the states associated with both games reactivated a large part of the visual cortex during sleep, similarly than during wakefulness (purple regions in Figure 4). However, the Face state significantly activated fusiform and occipital face-selective regions while the Maze state activated the para/hippocampal regions, consistent with task-specific neural replay during sleep.

Figure 4: Game-specific reactivation during N3 and subsequent memory. (A) In red, brain regions more activated for Face than Rest brain state during N3 sleep, and also recruited during the execution of the task at wake (blocks of Face vs. Rest; conjunction analysis). This network included, among other regions, early visual cortices and the fusiform and occipital face regions. In blue, brain regions more activated for Maze than Rest brain state during N3 sleep, and also recruited during the maze game (blocks of Maze vs. Rest; conjunction analysis). This network included, among others, early visual cortices, the hippocampus and parahippocampal cortex. In purple, overlap between those face and maze reactivation networks during sleep. **(B)** Correlation between activity in the face network (red regions in A) during N3 and memory performance for the face game. **(C)** Correlation between activity of the maze network (blue regions in A) during N3 and memory for the maze game. Grey dots for participants who won the face game; black dots for participants who won the maze game. FFA:

fusiform face area; paraH: parahippocampus; OFA: occipital face area. N=13. For memory performance measurements, see Supplementary Information.

We then extracted the mean beta values from each entire reactivation network, i.e. separately for Face and Maze states, and correlated these values with subsequent memory performance for each game. We observed significant positive correlations with memory performance related to the corresponding game (Spearman correlation; face: $\rho=0.66$, $P = 0.04$; Fig. 4B; maze: $\rho=0.73$, $P = 0.02$; Fig. 4C). While correlations do not ascertain causal relationships, this pattern of results converges with cell-recordings in animals¹ and human studies using targeted memory reactivation procedures²⁻⁴ to support a role for task-selective reactivations in memory consolidation processes. These findings also suggest that the first period of NREM sleep might be critical for the consolidation of declarative memory.

We report all these changes and results in the revised manuscript. We also toned down any causal interpretation of these data in the revised version of the manuscript.

Minor Points:

Comment #4

- Supplementary Table 1: Please provide the accuracy of the trained classifier during the Game Session for each individual subject. Do the individual differences in classifier accuracy during the Game session relate to the results during sleep? Furthermore, please add standard deviation to the average classification performance.

Response #4

We now show the individual values for the decoding accuracy in Figure S1 of the revised Supplementary Information file (see Figure below).

As suggested by the reviewer, we also tested whether the reactivation of the game states during sleep related to the classifier accuracy during the game playing session, and found no significant correlations (Reward: $R=-0.24$, $p=0.43$; no-Reward: $R=0.48$, $P=0.09$; Rest: $R=-0.31$, $p=0.31$; pre-R: $R=0.02$, $p=0.95$; pre-NR: $R=0.42$, $p=0.15$). These results suggest that the probability of reactivation during sleep did not relate to how accurately the classifier classified the game- or rest-related states during the training (game) session. We added this information in the Supplementary Information file page 12.

Comment #5

- Table 1: Please indicate statistical differences between the two experimental groups

Response #5

In the revised version of the manuscript, we do not report the data from two experimental groups separately anymore. The group distinction reported in the initial submission was actually a control manipulation (see our reply to the main comment #1 above). Yet, to address the reviewer's concern, we computed two ANOVAs, one on sleep duration and one on the percentage of total time spent in

the fMRI, with Sleep Stage (W, N1, N2 N3) as within-subjects factor and Group (face-win, maze-win) as between-subjects factor. We observed no main effects of Sleep Stage or Group, and no interaction for either ANOVA (all $p > 0.35$).

Comment #6

- Line 195: Please use some statistical procedure to confirm any overlap between task-related and reactivation-related brain activation. So far, the conclusion of “task-selectivity” appears not be supported by any statistical finding. In addition, the procedure of ROI selection is unclear. Which anatomical regions of the AAL atlas were taken and why? Did the authors have any a priori hypothesis on these anatomical regions? Why did they not take the results pattern during learning as one ROI? What do the authors exactly mean by small volume correction in Supplementary Table 4?

Response #6

We changed our approach to the investigation of task-related and reactivation-related activity. For more details about our new approach, please refer to our reply to Comment #3 above, and the corresponding text modification. In brief, to identify regions activated both during game playing and when the likelihood for game replay was high, we performed two statistical conjunction analyses between the contrasts Game playing (face or maze) > Rest during wakefulness and the reactivations during sleep (Face or Maze state vs. Rest state). Moreover, rather than selecting a priori ROIs (i.e. regions known to be involved in face perception or spatial navigation) for the correlation with memory performance, we extracted the beta values from the whole set of regions significantly activated in these conjunction analyses (see Supplementary Table 5 in the revision).

Small Volume Corrections (SVC) procedure in SPM applies the error correction to a predefined volume. This correction is used whenever an a-priori hypothesis does not apply to the entire brain, but only to certain brain areas. Standard whole-brain family-wise error (FWE) correction procedures assume that we are expecting and looking for effects (i.e., significant voxels) all around the brain. When this is not the case, and FWE is applied to the whole brain volume, the Type 1 error correction could be much more stringent than is required, meaning that the Type 2 error rate goes up (risk of not detecting an effect actually present). Therefore, specifying a priori ROIs for SVC allows a more sensitive test for the brain region of particular interest. In this study, we had the hypothesis that our face and maze tasks would recruit a specific set of brain regions involved in face perception and spatial navigation, respectively.

We used anatomical masks from Automated Anatomical Labeling (AAL). In this case, we reported brain activations for regions present at $p < 0.001$ uncorrected (whole-brain) and also significant at $p < 0.05$ FWE using SVC procedure at the voxel level for the peak of the region.

Reviewer #2

Comment #1

The paper is clearly written and the data potentially provide novel evidence bearing on mechanisms of replay during sleep and its role in memory consolidation. My main concern about the study is the sample size. The final analyses linking N3 activity (putative replay) with reward - the key findings of the paper - are performed on a very small sample of 7 and 6 participants in each condition. I recognize that this study is logistically very challenging, but I am concerned about basing conclusions on such a small sample. At the very least, it would be crucial to show individual data to determine how many of the 13 participants showed a difference between the reward and no-reward games during N3 sleep.

Response #1

Thanks you for this important comment, which we addressed as follows:

We would like to apologize for having evidently and regrettably misled this and the other reviewers by presenting our data as if we may have aimed at comparing the results from two independent, groups of participants. If this were indeed our aim, we would concur on the reviewers' concern about sample size. Note however that the data and methods reported in this study would actually allow looking at relatively small samples, as we discuss below. In fact, and as also stated in the initial submission, the goal of our study was to show that patterns of neural activity corresponding to one type of waking experience -- here playing one specific game -- were more likely to be spontaneously reactivated during NREM sleep if the game had been rewarded as compared to a non-rewarded game. We used a neural decoding approach to assess the likelihood of reactivation of distinct brain states specifically associated with playing the rewarded game and the non-rewarded game (and other states as well) during sleep. Because we could not exclude *a priori* that spontaneous patterns of brain activity during NREM sleep may intrinsically resemble those elicited when participants played one of the two games (i.e. face or maze), we carefully controlled for this possible confound by randomizing the reward factor across the games. This randomization allowed us to ascertain that greater reactivation of the brain state corresponding to one game during NREM sleep was attributable to the associated reward-value and not to the type of game itself. Therefore, our experimental design is a within-subjects design, including 13 participants in whom we compared a Reward (R) and No-Reward (NR) condition, and not 7 and 6 in each relevant experimental condition. To illustrate that this procedure successfully controlled for any spurious effect of the type of game, we initially presented separately the results from the participants who were assigned either game. We realize that this figure may have misled the reviewers and should have been presented as supplementary material. Therefore, we now present the main results for the whole sample of participants in Figure 3. In the original manuscript, we also reported an ANOVA with two within-subjects factors (Brain State and Sleep Stage) and one between-subjects factor (Type of Game), which may have further erroneously suggested that we aimed at comparing participants as a function of the specific game that was rewarded. We now describe the main analyses without this factor. We also show that 11 out of 13 participants exhibited higher likelihood values for the brain state corresponding to the rewarded game than the state corresponding to the non-rewarded game (see Figure 3 below), with one participant showing the reverse pattern and one showing very little reactivation.

Figure 3. Classification results (N=13). (A) Mean likelihood of Reward, No-Reward, Rest, pre-Reward (pre-R) and pre-No-Reward (pre-NR) brain states for each sleep stage, showing increased reactivation of the brain state associated with the rewarded game during N3. (B) Individual data for the Reward and No-Reward state during N3 sleep. (C) Correlation between time course of likelihood and low delta power (1-2 Hz) for the Reward and No-Reward states. Error bars represent SEM; dots represent individual data points.

Specifically, a repeated-measure ANOVA on likelihood measures with Brain state (Reward, No-Reward, Rest, pre-R, pre-NR) and Sleep stage (wake, N1, N2, N3) as within-subjects factors showed main effects of Brain state ($F_{(4,48)}=137.1$, $P < 0.001$) and an interaction between Brain state and Sleep stage ($F_{(12,144)} = 14.8$, $P < 0.001$). Post-hoc comparisons first showed that the Rest state predominated over the Reward and No-Reward states during wakefulness (Rest vs. Reward: $F_{(1,12)}=263.2$, $p < 0.001$, Rest vs. No-Reward: $F_{(1,12)}=515.8$, $p < 0.001$), while it was progressively less present from light to deeper sleep stages (main effect of sleep for the Rest state: $F_{(1,12)}=371.9$, $p < 0.001$). These results suggest a similarity between resting blocks during the game session and resting while awake during the sleep session. Critically, and confirming our main hypothesis, both game-related brain states had a relatively low likelihood to occur across all sleep stages (less than 0.20), and did not differ between Reward versus No-Reward states from wakefulness to N2 (R vs. NR for wake: $F_{(1,12)}=2.24$, $p=0.16$, N1: $F_{(1,12)}=0.18$, $p=0.67$, N2: $F_{(1,12)}=0.03$, $p=0.87$). By contrast, the Reward state differed from the No-Reward state during N3 sleep ($F_{(1,12)}=10.65$, $p=0.007$). The states corresponding to the preparation period before each game did not differ for the Reward and No-Reward states in any sleep stage (pre-R vs. pre-NR states for wake: $F_{(1,12)}=0.002$, $p=0.96$, N1: $F_{(1,12)}=0.37$, $P=0.56$, N2: $F_{(1,12)}=0.03$, $p=0.88$, N3: $F_{(1,12)}=1.42$, $P=0.26$).

All the corresponding changes in the presentation of the results are now implemented in the revised manuscript and Figure 3 has been amended accordingly.

Comment #2

Adding to this concern, in several places the analyses demonstrate a significant difference in one condition (e.g. N3) but not the other phases, but without a direct statistical comparison between them (e.g. line 163/Figure 3A, re significant interaction only in N3; line 215 re memory effects with the corresponding game, but not the other game).

Response #2

In the revision, we present the full ANOVA for the classification results with all 5 different states and all 4 different sleep stages, which now reads as follows:

“Specifically, a repeated-measure ANOVA on likelihood measures with Brain state (Reward, No-Reward, Rest, pre-R, pre-NR) and Sleep stage (wake, N1, N2, N3) as within-subjects factors showed main effects of Brain state ($F_{(4,48)}=137.1$, $P < 0.001$), and an interaction between Brain state and Sleep stage ($F_{(12,144)} = 14.8$, $P < 0.001$; Fig. 3A, Supplementary Table 3). Post-hoc comparisons first showed that the Rest state predominated over the Reward and No-Reward states during wakefulness (Rest vs. Reward: $F_{(1,12)}=263.2$, $p<0.001$, Rest vs. No-Reward: $F_{(1,12)}=515.8$, $p<0.001$), while it was progressively less represented from light to deeper sleep stages (main effect of sleep for the Rest state: $F_{(1,12)}=371.9$, $p<0.001$). These results suggest a similarity between resting blocks during the game session and resting while awake during the sleep session. Conversely, both game-related brain states had a relatively low likelihood of being present across all sleep stages (less than 0.20), and did not differ between Reward versus No-Reward states from wakefulness to N2 (R vs. NR for W: $F_{(1,12)}=2.24$, $p=0.16$, N1: $F_{(1,12)}=0.18$, $p=0.67$, N2: $F_{(1,12)}=0.03$, $p=0.87$). By contrast, and confirming our main hypothesis, the Reward state differed from the No-Reward state during N3 sleep ($F_{(1,12)}=10.65$, $p=0.007$). The states corresponding to the preparation period before each game did not differ for the rewarded and non-rewarded games in any sleep stage (pre-R vs. pre-NR states for W: $F_{(1,12)}=0.002$, $p=0.96$, N1: $F_{(1,12)}=0.37$, $P=0.56$, N2: $F_{(1,12)}=0.03$, $p=0.88$, N3: $F_{(1,12)}=1.42$, $P=0.26$).”

These results are now included in the Results section of the revised manuscript.

Comment #3

Figure 3CD: it would be helpful to visualize which of these 13 individual data points corresponds to Ss who won the maze vs. the face game.

Response #3

The figure has been modified according to the new analyses performed to assess task-related reactivation and subsequent memory performance (see also our reply to Comment #3 of Reviewer #1). Participants who won the face or maze are now displayed in different colors (see Figure 4 above).

Reviewer #3

Comment #1

The authors try to link the reactivation they observe with the hippocampal replays. Nonetheless, they do not point out at the major differences between these two phenomena. Hippocampal replay are short (few hundred milliseconds) and localized in hippocampal regions whereas, here, they observed the reactivation of patterns of brain activity across several cortical regions and at much larger time-scales (since they used the BOLD signal). It is quite hard to link these two phenomena. Indeed, the condensed nature of hippocampal replays allow them to be embedded in sleep oscillations (ripples, spindles and slow-waves) which is deemed crucial for the consolidation of the associated memory. If the representation of the games were here replayed during sleep with positive consequences on subsequent memory, as the authors claim, this must be done through very different mechanisms than as for hippocampal replays. This should be discussed in the manuscript.

Response #1

We thank the Reviewer for these suggestions. Indeed, the Reviewer rightly pointed out that we primarily reported reactivations across cortical regions, while we suggested that these reactivations should be coordinated by the hippocampus, and implicate reward regions for reward-related memory traces. To achieve a more comprehensive description of the mechanisms involved, in particular in the hippocampus, and of the possible recruitment of dopaminergic reward-related regions, we performed new analyses focusing on the activity of two a-priori ROIs -- the hippocampus and the VTA -- during the sleep session. We observed that the time courses of the hippocampus and the VTA correlated with the Reward state (more than with the No-reward state; Figure 5 below), which itself correlated with slow wave activity (see Figure 3, reply to Comment #4 below). These findings suggest that, during NREM sleep, the spontaneous reactivation of cortical representations of rewarded information was not only temporally aligned with slow oscillations but also with increased hippocampal and VTA recruitment. We thus provide strong additional support to the hypothesis that the hippocampus and the VTA are involved in the reactivation of rewarded memories during sleep.

Figure 5: Reward-related reactivation in hippocampal and ventral tegmental area (VTA) during sleep (N=13). (A) Anatomical masks used as regions of interest (in green). (B) Mean correlation values

for all participants between the time courses of the regions of interest and the different brain states. Asterisks indicate $P < 0.01$ for the Reward vs. No-Reward comparison.

We amended the text as follows (last paragraph of the revised Results section) and included Figure 5 as an additional figure in the main text:

“Because during game playing, both games engaged memory regions, and because the reward status of the games was not included in the training dataset (rewarded block not included), the classifier could not distinguish the games on the basis of activity across memory and/or reward networks. Yet, based on previous animal and human studies^{1,5}, we expected that the reactivation of the Reward state may be associated with an increased recruitment of memory and reward regions. We thus tested whether the hippocampus and ventral tegmental area (VTA), known to be critically involved in episodic memory and reward respectively, were preferentially engaged whenever the rewarded state was detected during sleep. To this end, we extracted the time courses of activity from two a priori anatomically-defined regions, i.e. the bilateral hippocampus (using the AAL atlas, Fig. 5A) and bilateral VTA (using a manually defined ROI on proton-density images from an independent sample of 19 participants). We then assessed the correlation between these values and the 5 different brain states decoded during the sleep session, and performed two ANOVAs on the rho values, with the 5 states as a within-subjects factor (Fig. 5B). For the hippocampus, we observed a main effect of State ($F(4,48)=80.19$, $p < 0.001$). Post-hoc analyses revealed that the time course of hippocampal activity correlated significantly more with the Reward state than the other states (vs. No-Reward state: $F(1,12)=15.21$, $p=0.002$; vs. Rest state: $F(1,12)=95.15$, $p < 0.001$; vs. pre-R state: $F(1,12)=80.98$, $p < 0.001$; vs. pre-NR: $F(1,12)=80.50$, $p < 0.001$). Similarly, for the VTA, we observed a main effect of State ($F(4,48)=21.0$, $p < 0.001$) and the time course of VTA activity correlated significantly more with the Reward state than the others ones (vs. No-Reward state: $F(1,12)=7.21$, $p=0.02$; vs. Rest state: $F(1,12)=25.37$, $p < 0.001$; vs. pre-R state: $F(1,12)=22.73$, $p < 0.001$; vs. pre-NR: $F(1,12)=21.89$, $p < 0.001$).”

Regarding the question whether using fMRI technology allows detecting brief neural events, this has been extensively discussed in previous studies and proven valid. Typically, fMRI can detect changes in activity in responses to short external stimulations of a few milliseconds, corresponding to changes in evoked-related neural responses within the tens of milliseconds range (as detected by EEG or LFP). This is for example true for fMRI responses to subliminal visual stimuli (using brief and masked stimulus presentation⁶, or even attentional modulation of brief visual stimuli, which in the EEG affects the earliest ERP component C1^{7,8}). Using a combination of hippocampal local field potential recordings and fMRI, Ramirez-Villegas et al. (2015) were able to detect differential brain-wide modulation of fMRI activity for distinct sharp-wave-ripple types, in the monkey brain⁹. In humans, Bergmann et al.¹⁰ demonstrated that category-specific cortical reactivation during sleep after learning face-scene associations occurred in temporal synchrony with spindle events and was tuned by ongoing variations in spindle amplitude (see also¹¹).

Based on these and other previous studies, we suggest that increased BOLD response in the hippocampus, and in task- and reward-related brain regions (see also our reply to Comment #2 below) indexes a spontaneous reactivation phenomenon, which may most plausibly relate to hippocampal sharp-wave ripples. The strength of our approach is to demonstrate, for the first time to our knowledge, reward-biased memory reprocessing during N3 in humans, involving distributed large-scale neural networks. Yet, the temporal resolution of fMRI does not allow testing for the

reactivation of temporally-organized individual neurons's activity (such as during ripples), and should therefore be considered as a complementary approach to single-cell recordings. Similarly, the detailed temporal relationship (e.g. at the 10-100 ms resolution) between hippocampal activity and transient changes in EEG activity cannot be assessed by the EEG-fMRI approach used in the present study. Yet, future studies using fast fMRI techniques and/or ultra-high-field systems may overcome this limitation^{12,13}.

We added the following sentence in the Discussion section of the revised manuscript:

“The temporal resolution of fMRI does not allow to test for the reactivation of temporally-organized individual neurons' activity (such as during ripples), and should therefore considered as a complementary approach to single-cell recordings.”

Comment #2

The authors claim that they 'uncover[ed] a neural mechanism whereby the most valuable elements of our lives become engraved on our memory while we sleep' (l. 35). This study actually falls short at providing such mechanism. The authors do show that representations associated with rewards are more likely to be reactivated in sleep, but they don't show or propose any neural mechanisms explaining how this is done. It would have been interesting to see how the re-activations occurring during N3 relate to the reward-circuits and what triggers the reactivation of the winning game.

Response #2

As proposed by the reviewer, we performed a new analysis using as a region of interest the VTA, a key region of the reward system, and found that activity in the VTA increased whenever the likelihood of the Reward state also increased. We modified the result section (see also our response to Comment #1 above), and also added the following sentences in the Discussion section of the revised manuscript.

“To specifically test for the implication of the reward system during reward-biased reactivation in sleep, we used a region of interest approach and observed that activity of the VTA increased during sleep whenever the likelihood of the Reward state was high.”

Comment #3

I find the authors' choices to train and validate their classifier surprising. They indeed trained the classifier on 5 classes that partly overlapped. Indeed, the pre-face and pre-maze states seem quite close to the Rest state. In addition, the pre-face and pre-maze states comprised 8*3s of data whereas the face and maze states comprised 8*60s of data and the Rest state 16*90s! It seems that this choice of unbalanced and overlapping classes can only add noise to the fitting of the classifier. Accordingly, classifier's validation (Fig. S1) shows that the pre-face and pre-maze classes were not correctly decoded. I would advise the authors to fit the classifier on three classes only (face, maze and Rest). As a side note, the authors should quantify the performance of the classifier for each class and overall using F1-scores or ROC-curves.

Response #3

We used 5 different states to avoid that the classifier's output would be forced to assign each fMRI image to only 2 or 3 states. Indeed, our reasoning was that using only two games-related and one rest-related state may introduce unwanted biases, such as the classifier favoring the rest state whenever activation pattern was very dissimilar to reactivation patterns. This issue was also pointed out by Reviewer 1 (see also our reply to Comment #2, Reviewer 1). We concur with the current reviewer that low accuracy for the pre-face and pre-maze states is most probably due to their short duration (i.e. few data points) during the game session relative to the Face, Maze and Rest states. Although not supported by the final results, we initially thought that such pre-games states, during which participants anticipate (i.e. mentally prepare for) one or the other upcoming game, may also occur during wakefulness prior to sleep, like for the Rest state. Concerning the performance of the classifier for each class, we now show all individual data for the 5 classes in Figure S1 of the revised manuscript (see also our reply to Comment #4 from Reviewer 1). Please note that computing ROC-curves to visualize all possible trade-offs in terms of TPR and FPR is not a trivial procedure for a multi-class classifier such as the one that we used here and, as the reviewer seems to acknowledge, this may not be a critical verification, especially given the additional information that we report in the revised version of the manuscript. We would of course perform this analysis whether required by the reviewer.

Comment #4

The authors used a between-subjects design in which they crucially tested the significance of a triple interaction between two within-subjects factors (Brain-state and Sleep-stage) and a between-subject factor (Group). Surprisingly, they only have 6 and 7 participants in the two Groups. The study seems thus quite underpowered. The authors should check that (i) their data meet the requirements of the repeated measure ANOVA despite the small sample size, (ii) they have sufficient power to check for the triple interactions. If this is not the case, the authors will have to modify their analyses and/or acquire new data after estimating the sample-size needed.

Response #4

This is a very important point, which prompted us to clarify a key feature of our experimental design as follows:

We would like to apologize for having evidently and regrettably misled this and the other reviewers by presenting our data as if we may have aimed at comparing the results from two independent groups of participants. If this were indeed our aim, we would concur on the reviewers' concern about sample size. Note however that the data and methods reported in this study would actually allow looking at relatively small samples, as we discuss below. In fact, and as also stated in the initial submission, the goal of our study was to show that patterns of neural activity corresponding to one type of waking experience -- here playing one specific game -- were more likely to be spontaneously reactivated during NREM sleep if the game had been rewarded as compared to a non-rewarded game. We used a neural decoding approach to assess the likelihood of reactivation of distinct brain states specifically associated with playing the rewarded game and the non-rewarded game (and other states as well) during sleep. Because we could not exclude *a priori* that spontaneous patterns of brain activity during NREM sleep may intrinsically resemble those elicited when participants played one of the two games (i.e. face or maze), we carefully controlled for this possible confound by randomizing the reward factor across the games. This randomization allowed us to ascertain that

greater reactivation of the brain state corresponding to one game during NREM sleep was attributable to the associated reward-value and not to the type of game itself. Therefore, our experimental design is a within-subjects design, including 13 participants in whom we compared a Reward (R) and No-Reward (NR) condition, and not 7 and 6 in each relevant experimental condition. To illustrate that this procedure successfully controlled for any spurious effect of the type of game, we initially presented separately the results from the participants who were assigned either game. We realize that this figure may have misled the reviewers and should have been presented as supplementary material. Therefore, we now present the main results for the whole sample of participants in Figure 3. In the original manuscript, we also reported an ANOVA with two within-subjects factors (Brain State and Sleep Stage) and one between-subjects factor (Type of Game), which may have further erroneously suggested that we aimed at comparing participants as a function of the specific game that was rewarded. We now describe the main analyses without this factor. We also show that 11 out of 13 participants exhibited higher likelihood values for the brain state corresponding to the rewarded game than the state corresponding to the non-rewarded game (see Figure 3 below), with one participant showing the reverse pattern and one showing very little reactivation.

Figure 3. Classification results (N=13). (A) Mean likelihood of Reward, No-Reward, Rest, pre-Reward (pre-R) and pre-No-Reward (pre-NR) brain states for each sleep stage, showing increased reactivation of the brain state associated with the rewarded game during N3. (B) Individual data for the Reward and No-Reward state during N3 sleep. (C) Correlation between time course of likelihood and low delta power (1-2 Hz) for the Reward and No-Reward states. Error bars represent SEM; dots represent individual data points.

Specifically, a repeated-measure ANOVA on likelihood measures with Brain state (Reward, No-Reward, Rest, pre-R, pre-NR) and Sleep stage (wake, N1, N2, N3) as within-subjects factors showed main effects of Brain state ($F_{(4,48)}=137.1, P < 0.001$) and an interaction between Brain state and Sleep stage ($F_{(12,144)} = 14.8, P < 0.001$). Post-hoc comparisons first showed that the Rest state predominated

over the Reward and No-Reward states during wakefulness (Rest vs. Reward: $F_{(1,12)}=263.2$, $p<0.001$, Rest vs. No-Reward: $F_{(1,12)}=515.8$, $p<0.001$), while it was progressively less present from light to deeper sleep stages (main effect of sleep for the Rest state: $F_{(1,12)}=371.9$, $p<0.001$). These results suggest a similarity between resting blocks during the game session and resting while awake during the sleep session. Critically, and confirming our main hypothesis, both game-related brain states had a relatively low likelihood to occur across all sleep stages (less than 0.20), and did not differ between Reward versus No-Reward states from wakefulness to N2 (R vs. NR for wake: $F_{(1,12)}=2.24$, $p=0.16$, N1: $F_{(1,12)}=0.18$, $p=0.67$, N2: $F_{(1,12)}=0.03$, $p=0.87$). By contrast, the Reward state differed from the No-Reward state during N3 sleep ($F_{(1,12)}=10.65$, $p=0.007$). The states corresponding to the preparation period before each game did not differ for the rewarded and non-rewarded games in any sleep stage (pre-R vs. pre-NR states for wake: $F_{(1,12)}=0.002$, $p=0.96$, N1: $F_{(1,12)}=0.37$, $P=0.56$, N2: $F_{(1,12)}=0.03$, $p=0.88$, N3: $F_{(1,12)}=1.42$, $P=0.26$).

All the corresponding changes in the presentation of the results are now implemented in the revised manuscript and Figure 3 has been amended accordingly.

Comment #5

Given the small sample-size in the two groups ($n=6$ and 7), the authors should avoid the use of bar-plots (e.g. in Figure 3), especially since they did not define the nature of the error-bars used. The authors should plot the individual data. The authors should also focus on one sample (i.e. the subjects with enough N3) for clarity and should, in any case, always mention the sample-size for all figures and statistical tests.

Response #5

We thank you for this comment. In the revised Figure 3 (see above), we now plot the individual data for each condition, and indicate that the error bars represent SEM. We also specify the number of participants included in the analysis for each figure and each table.

Comment #6

The authors investigated the relationship between game-related re-activations and delta activity. However, from the different regions showing differential activations between the face (or maze) state and Rest (see ST4), they report only the results for the regions leading to significant results (OFA and hippocampus, Fig. 3 C-D), which is quite close to cherry-picking the results. Since they tested many more regions and frequency ranges (l. 193-194), they should correct their results for multiple comparisons. I am not sure the results would hold with such constrains.

Response #6

We agree with the Reviewer that, although valid, selecting specific regions of interest while ignoring the whole network of regions involved in reactivation may not be the optimal procedure. We therefore changed our approach and extracted activity from all brain regions commonly activated for the contrast face (or maze) vs. rest during both the awake game session and N3 sleep (i.e. Face or Maze state vs. Rest state) using conjunction analyses (new Figure 4 below). We could confirm that the occurrence of the states related to each game during deep sleep activated a subset of regions involved when playing at the corresponding game while awake (Figure 4 in red and blue, see also

revised Supplementary Table 5). During N3, the states associated with both games reactivated a large part of the visual cortex during sleep, similarly than during wakefulness (purple regions in Figure 4). However, the Face state significantly activated fusiform and occipital face-selective regions while the Maze state activated the para/hippocampal regions, consistent with task-specific neural replay during sleep.

Figure 4: Game-specific reactivation during N3 and subsequent memory. (A) In red, brain regions more activated for Face than Rest brain state during N3 sleep, and also recruited during the execution of the task at wake (blocks of Face vs. Rest; conjunction analysis). This network included, among other regions, early visual cortices and the fusiform and occipital face regions. In blue, brain regions more activated for Maze than Rest brain state during N3 sleep, and also recruited during the maze game (blocks of Maze vs. Rest; conjunction analysis). This network included, among others, early visual cortices, the hippocampus and parahippocampal cortex. In purple, overlap between those face and maze reactivation networks during sleep. **(B)** Correlation between activity in the face network (red regions in A) during N3 and memory performance for the face game. **(C)** Correlation between activity of the maze network (blue regions in A) during N3 and memory for the maze game. Grey dots for participants who won the face game; black dots for participants who won the maze game. FFA: fusiform face area; paraH: parahippocampus; OFA: occipital face area. $N=13$. For memory performance measurements, see Supplementary Information.

This new approach and the ensuing results are included in the main text and in the Supplementary Information file of the revised manuscript.

Comment #7

In the following analysis (l. 195-203), the authors seek to check which regions were involved during the N3 re-activations: ‘We thus included the likelihood of each brain state for each fMRI volume during N3 sleep as a regressor in a whole-brain regression analysis’ (l. 197). They observed significant activations within the task-related ROI. This is quite trivial and akin to double-dipping. Indeed, the

classifier was trained on specific ROIs. The likelihood for the MAZE pattern, for example, would therefore increase when the MAZE ROI get activated. Thus, finding activations during N3 that match the initial ROI could just reflect how the classifier was trained. In my opinion, it does not allow to conclude on any 'task-selectivity' (l. 202) or region-specificity (l. 196).

Response #7

We respectfully disagree with the reviewer's opinion and instead would like to stress that this analysis is not as trivial as it may seem. In particular, we may not exclude that reactivation of the game-states during sleep would also (and potentially mainly) recruit brain regions outside of the selected ROIs, notably because we trained and applied the classifier across two distinct states of the brain (i.e. wakefulness and sleep). We therefore used the time course of each state during sleep without specifying any region of interest. In other words, this analysis can be seen as complementary of or a control of the results of the classifier, showing that spontaneous reactivations during sleep recruited a similar set of brain regions as during wakefulness. Of course, the main result of the present work still pertains and fully depends on the classification outcome: i.e. increased likelihood of the Reward state to occur during N3. See also our reply to Comment #6 above.

Comment #8

2.6. In the main text, most of the statistical tests are not properly described (test name, sample-size, effect-size...), which makes it impossible to assess the reliability of the statistical analyses.

Response #8

We now report all effects size for each result, and clarify the statistical tests used.

Minor comments:

Comment #9

1. I find the definition of the REST contrast and ROI puzzling: 'the two latter contrasts were used to reveal regions whose activity was suppressed immediately following each of the games relative to later periods of rest' (SI: l. 261). Why focusing on regions suppressed after each game? I would expect a REST-state to cover mostly regions within the Default-Mode Network. How do the ROI for the REST state overlap with the DMN? This would be important to understand why the REST state tends to disappear after sleep onset.

Response #9

In a previous paper¹⁴, we demonstrated that activity evolves across time in a resting period of 90 sec after the presentation of emotional or neutral videos. In particular, we found that those regions whose activity was significantly suppressed immediately after the end of the videos (first 30sec relative to subsequent 30sec temporal bins) largely overlapped with those activated during video watching. As expected, these contrasts (for face and for maze) mostly engaged task-selective regions, very little overlap with the DMN (see Supplementary Table 1). We combined increased activations during game playing and decreased activations for first compared to last Rest temporal bins to make

sure that we would get most of the network involved in each of the games. We have now further clarified this point in the revised Supplementary Information file.

Comment #10

2. In the manuscript, the authors should draw a clear distinction between the notion of replay (as seen in rodents: a clear repetition of a sequence of firing condensed in time) and the reactivation of patterns of brain activity similar to the game-related activation as observed here.

Response #10

We amended the manuscript so that we now use “replay” only for cell-recording studies in rodents and “reactivation” for the reemergence of patterns of brain activity similar to the game-related activation. We thus make it clearer that these two terms do correspond to different types of experimental measurements.

Comment#11

3. Along the same line, the authors wrote that ‘This finding may implicate a coordinated replay of hippocampal and striatal regions during sleep, as previously observed after a spatial rewarded task in rodents [16]’ (l. 243). However, the reference cited here (Lansik et al. 2009) showed a much different form of replay than the reactivation evidence here (short replay associated to hippocampal ripples). In addition, the authors did not show here any form of coordination between hippocampal and striatal regions.

Response #11

As proposed by the Reviewer in his/her Comments #1 and #2, we computed new analyses focusing on hippocampus and reward related regions. We now report these new analyses along with a new Figure 5 (see also our Response #1 and #2 above).

Comment #12

4. Would not it be preferable to exclude the last blocks from the classifier since these blocks could be contaminated by activation associated to the fact of winning?

Response #12

The data from the last block when the participant won at one or the other game was not included in the classification. We thank you for raising this important point, which we now clarify in the Supplementary Information file as follows:

“Please note that, as for all main contrasts (see above), the extracted time-courses of fMRI activity did not include the data from the block during which participants won one of the games (and after which the game ended).”

Comment #13

5. Why using a repeated measure ANOVA (with strong assumptions on the equality of variances hard to meet with small sample sizes) and not non-parametric method?

Response #13

Please note that all analyses are now performed with a minimum sample size of 13 participants.

To directly address the Reviewer’s concern, we also computed a permutation repeated measures ANOVA with 5 states and 4 sleep stages as conditions. The interaction between these two factors is significant ($p \approx 0.0002$, 5000 permutations) and the non-parametric comparison between Reward and no-Reward states during N3 is also significant at $p \approx 0.008$ (permutation paired t-test, 500 random sign flips on the difference scores of Reward and no-reward in N3).

Comment #14

6. Why was memory performance z-scored? Could the authors also provide an analysis of the raw memory performance (in particular were the participants above chance-level two days after the game session?)?

Response #14

For the memory of the face game, the maximum number of point was 54 for 18 faces (3 points for each correct location, 1 point for each correct column, 0.5 point for each correct row; 3 points). For a grid of 3 rows by 6 columns, chance level would thus be at 7.5. As can be seen in the table below, performance was above chance in both the face win ($t(1,13)=5.12$, $p<0.001$) and maze win groups ($t(1,13)=-10.2$, $p<0.001$).

For the maze game, the best possible performance was zero, when participants reached the starting location, and the worst performance was 50 (in arbitrary distance on the map for the farthest location). Our subjects performed between 0 and 34 distance units from the starting location after 30 sec. Distances for the maze game were inversed; so that larger score indexed better performance. For our navigation task, no chance-level performance can validly be computed.

	Face memory	Maze memory
Face win participants	13.43 ± 4.49	14.57 ± 13.06
Maze win participants	16.75 ± 5.91	17.67 ± 11.54

As suggested by the reviewer, for the correlation analysis with brain activity, we now use the raw data (see Figure 4BC). We also tested whether winning at one game favors the consolidation of this game or not. To be able to compare memory performance of the rewarded vs. non-rewarded game (face or maze) in the same ANOVA, we had to Z-score them to combine face and maze memory performance. These points are clarified in the revised version of the Supplementary Information file.

Comment #15

7. Why did the authors divide the delta-band in two? Why choosing a lower boundary at 1Hz and not 0.5Hz, a more classical value?

Response #16

We wanted to divide the frequency band as precisely as possible. According to several papers¹⁵⁻¹⁷, delta band can be subdivided in slow (0.7-2 Hz) and fast (2-4Hz) delta, with the power from the two delta bands only moderately correlating with each other. Fast delta was reported to increase mainly from drowsiness to slow wave sleep (SWS) onset, whereas slow delta increased markedly during SWS¹⁵. It is therefore interesting to note that only slow delta correlated significantly with the rewarded state. We now clarify this point in the Supplementary Information file as follows:

“We subdivided the delta band in two subcomponents (low and fast delta) to take into account possible functional distinctions between both subcomponents¹⁵.”

Comment #17

8. ‘No overall difference’ (l. 207) Could the authors specify the tests performed and the results obtained?

Response #17

We now clarify this result in the revised Supplemental Information file as follows:

“A repeated-measures ANOVA on these values with Memory task (face, maze) as within-subjects factor and Won game (participants who won at the face game and those who won at the maze game) as between-subjects factor showed no significant effect of Memory task ($F_{(1,11)}=0.007$, $p=0.93$), Won game ($F_{(1,11)}=0.21$, $p=0.65$), or interaction ($F_{(1,11)}=1.30$, $p=0.28$).”

Comment #18

9. The authors wrote: ‘The present study adds unprecedented experimental evidence for a close resemblance between neural activity patterns associated with specific behaviors trained during wakefulness and those spontaneously generated during sleep’ (l. 233). The authors did not really show a ‘close resemblance’. How close the reactivation is from the original activation is assessed relatively to other states. It could be quite far and still discriminate the different states. Similarly, the authors should avoid terms as ‘strongly regulated’ as it is difficult to assess the strength of the results with such a small sample-size.

Response #18

In the revision, we report new results from a conjunction analysis between the brain regions activated during the games at wake and those showing state-related reactivation during sleep. Unlike the visual overlap from the previous version of the manuscript, this new methodological approach is statistically robust. Furthermore, it does not require the selection of a priori ROIs (see also our response to Comment #6 above).

We removed the term “strongly” regulated in the discussion.

Comment #19

10. 'further confirming that this reactivation implicates those same regions that were recruited during wakefulness' (l. 236). See also major comment 2.5, this cannot be concluded from the analysis performed.

Response #19

As explained in our responses to Comments #6 and #18, we do not rely on a mere visual inspection anymore, but now provide statistical evidence for the differential reactivation of patterns of brain activity seen during playing the face or maze game, respectively.

Comment #20

11. In the last paragraph, the authors advocate for separate effects of REM and NREM sleep for the reactivation of negative or positive memories respectively. This is very speculative since (i) such contrast has not been directly tested, neither here nor in the work cited, (ii) as the authors mention, the subjects may have experience negative emotions as well, associated to the losing game.

Response #20

We agree that it is speculative; this is why we introduced this point as a potential limitation of our study, at the end of the discussion section.

Comment #21

12. Regarding Table ST1, could the authors perform an analysis of potential differences between the two groups (FACE-win, MAZE-win)?

Response #21

In the new version of the manuscript, we do not separate participants into distinct groups. Yet, to address the reviewer's comment, please find below the requested analysis. As can be seen, and although performed on small samples of participants (face win: n=7; maze win: n=6), participants from each group did not differ much while playing the games (one cluster of 7 voxels and one of 1 voxel surviving a threshold of $p < 0.001$ uncorrected).

Table R1: Main clusters of activation during the game session for the face win and maze win groups

	Nbr voxels	X (mm)	Y (mm)	Z (mm)	Z score
Face > maze game for face win vs. maze win participants (or maze > face game, maze win vs. face win)					
Superior occipital cortex	7	12	-91	34	3.89
Face > maze game for maze win vs. face win participants (or maze > face game, face win vs. maze win)					
-					
Face-rest1 < face-rest3 for face win vs. maze win participants					
-					
Face-rest1 < face-rest3 for maze win vs. face win participants					
-					

Maze-rest1 < maze-rest3 for face win vs. maze win participants					
Temporal pole	1	-36	11	-29	3.47
Maze-rest1 < maze-rest3 for maze win vs. face win participants					
-					

Comment #22

13. Could the authors provide the scores of participants in the different questionnaires made (SI: I. 41-48)?

Response #22

As recommended by the Reviewer, we have added the scores for the different questionnaires in the Supplementary Information file as follows:

“They were not depressed as assessed by the Beck Depression Inventory¹⁸ (mean ± SD: 1.7 ± 2.0), and had low anxiety levels as assessed by the STAI-T¹⁹ (31.8 ± 5.8). None of the participants suffered from excessive daytime sleepiness as assessed by the Epworth Sleepiness Scale²⁰ (5.6 ± 3.0) or sleep disturbances as determined by the Pittsburgh Sleep Quality Index Questionnaire²¹ (3.1 ± 2.2). Sensitivity to Punishment and Sensitivity to Reward Questionnaire²² established that none of the participants had extreme sensitivity to reward (37.0 ± 7.7) or punishment(32.5 ± 5.4), nor did they suffer from excessive impulsivity as assessed by the UPPS Impulsive Behavior Scale^{23,24} (90.1 ± 9.6).”

Comment #23

14. ‘participants had to believe that they won’ (SI: I. 67). Was this checked? How?

Response #23

We performed extensive piloting of both tasks to achieve this goal. Thus, during piloting we explicitly asked participants if they thought that they won by themselves (because of their own skills, ability, intelligence), and then if they suspected that we may have rigged the games. In the version that we used, all pilot participants (N=8) tested in the final version did report that they managed to find the correct face (in the face game) or the exit (in the maze game) by themselves and did not think that the games were rigged. We did not explicitly ask the real participants in the present study whether they noticed that the games were rigged. Yet, none of the participant mentioned that he/she noticed anything like a fake manipulation of the games during a general debriefing at the very end of the experiment. According to the debriefing, participants were satisfied by their performance at the Rewarded game (6.0 ± 1.4 on a 10 point scale from 0: very unsatisfied to 10: very satisfied) and much less for the Non rewarded game (3.5 ± 1.4 on the same scale, paired t-test: t(1,13)=5.89, p<0.001)), as we now report in the revised manuscript. Moreover, the imaging results in sleep strongly suggest that our manipulation was successful in associating a reward value to the game that was won, both regarding the increased reactivation of the brain state associated with the rewarded game, and of the simultaneous increased activation of the VTA (see also our replies to Comments #1 and #2). As a reminder, we built the games so that the participants could not realize that it was not possible to win at both games. Two participants did not win at any of the games and were removed from the analyses.

Comment #24

15. 'EEG data analyses': what is 'a sliding average of 21 averages' (SI: l. 166)?

Response #24

Gradient artifacts were removed offline using BrainVision Analyzer software (Brain Products Inc, Munich, Germany). Here is the explanation of the Brain vision Analyser software manual: "The Use Sliding Average Calculation option allows you to address the problem of combined EEG-fMRI measurement, namely that the he scanner artifacts may sometimes be greatly modified by even slight movements of the test subject's head in the scanner. This would substantially reduce the quality of a template calculated across all intervals. A solution to this problem is to calculate the template by means of a sliding average of a selected number of intervals. A separate template is calculated for each interval to correct the interval. In the Total Number of Intervals in Template text box, enter the number of intervals that are to be used for the calculation of the correction template."

In our analysis, we calculated a separated template for each interval, and we used 21 intervals for the calculation of the template.

Comment #25

16. It is weird for the authors to mention that they designed a classifier with the expectation that it would fail to classify: 'classification accuracy was high for the FACE, MAZE, and REST states (all > 0.76), but low for the pre-FACE and pre-MAZE states, as expected' (SI: l. 267-269).

Response #25

You are absolutely right, and we apologize for this unfortunate formulation. We initially did not have such an expectation (see also our reply to Comment #3 above), but only subsequently realized that the short duration of the pre-game period may explain the low accuracy for classification. We therefore have removed the term "as expected" from the revised version of the manuscript.

Comment #26

17. Table ST3: were correlations for sigma band significant?

Response #26

Here is the new description of the ANOVA in the main text:

"The resulting correlation values entered an ANOVA with Brain state (Reward, No-Reward, pre-R, pre-NR, Rest) and Frequency band (low delta, 1-2 Hz; high delta, 2-4 Hz; theta, 4-7 Hz; alpha, 8-10 Hz; sigma, 12-14 Hz; beta, 15-25 Hz) as within-subjects factors. We observed a main effect of Brain state ($F_{(4,48)}=8.51$, $p<0.001$), a main effect of Frequency band ($F_{(5,60)}=13.59$, $p<0.001$), and an interaction between both factors ($F_{(20, 240)}=15.95$, $p<0.001$). More specifically, a direct comparison between Reward and No-Reward states showed a significant difference ($F_{(1,12)}=5.54$, $p=0.03$) due to the positive correlation between low delta activity and the strength of the reactivation of the brain state associated with the successful game (Fig. 3C, Supplementary Table 4). No such interaction was found when exploring correlations of game-related states with other frequency bands (Supplementary Table 4)."

For clarity, please find here the comparison between Reward vs. No-Reward states, for each frequency band: low delta: $F(1,12)=5.54$, $p=0.03$; high delta: $F(1,12)=3.51$, $p=0.09$; theta: $F(1,12)=1.33$, $p=0.27$; alpha: $F(1,12)=0.13$, $p=0.75$; sigma: $F(1,12)=0.004$, $p=0.95$; beta: $F(1,12)=1.68$, $p=0.22$).

References

- 1 Lansink, C. S., Goltstein, P. M., Lankelma, J. V., McNaughton, B. L. & Pennartz, C. M. Hippocampus leads ventral striatum in replay of place-reward information. *PLoS biology* **7**, (2009).
- 2 Rasch, B., Buchel, C., Gais, S. & Born, J. Odor cues during slow-wave sleep prompt declarative memory consolidation. *Science* **315**, 1426-1429, (2007).
- 3 Rudoy, J. D., Voss, J. L., Westerberg, C. E. & Paller, K. A. Strengthening individual memories by reactivating them during sleep. *Science* **326**, 1079, (2009).
- 4 Shanahan, L. K., Gjorgieva, E., Paller, K. A., Kahnt, T. & Gottfried, J. A. Odor-evoked category reactivation in human ventromedial prefrontal cortex during sleep promotes memory consolidation. *eLife* **7**, (2018).
- 5 Igloi, K., Gaggioni, G., Sterpenich, V. & Schwartz, S. A nap to recap or how reward regulates hippocampal-prefrontal memory networks during daytime sleep in humans. *eLife* **4**, (2015).
- 6 Kouider, S., Barbot, A., Madsen, K. H., Lehericy, S. & Summerfield, C. Task relevance differentially shapes ventral visual stream sensitivity to visible and invisible faces. *Neuroscience of consciousness* **2016**, niw021, (2016).
- 7 Rauss, K. S., Pourtois, G., Vuilleumier, P. & Schwartz, S. Attentional load modifies early activity in human primary visual cortex. *Hum Brain Mapp* **30**, 1723-1733, (2009).
- 8 Schwartz, S. *et al.* Attentional load and sensory competition in human vision: modulation of fMRI responses by load at fixation during task-irrelevant stimulation in the peripheral visual field. *Cerebral cortex* **15**, 770-786, (2005).
- 9 Ramirez-Villegas, J. F., Logothetis, N. K. & Besserve, M. Diversity of sharp-wave-ripple LFP signatures reveals differentiated brain-wide dynamical events. *Proc Natl Acad Sci U S A* **112**, E6379-6387, (2015).
- 10 Bergmann, T. O., Molle, M., Diedrichs, J., Born, J. & Siebner, H. R. Sleep spindle-related reactivation of category-specific cortical regions after learning face-scene associations. *Neuroimage* **59**, 2733-2742, (2012).
- 11 Jegou, A. *et al.* Cortical reactivations during sleep spindles following declarative learning. *Neuroimage* **195**, 104-112, (2019).
- 12 Lewis, L. D., Setsompop, K., Rosen, B. R. & Polimeni, J. R. Fast fMRI can detect oscillatory neural activity in humans. *Proc Natl Acad Sci U S A* **113**, E6679-E6685, (2016).
- 13 Fruhholz, S., Trost, W., Grandjean, D. & Belin, P. Neural oscillations in human auditory cortex revealed by fast fMRI during auditory perception. *Neuroimage* **207**, 116401, (2020).
- 14 Eryilmaz, H., Van De Ville, D., Schwartz, S. & Vuilleumier, P. Impact of transient emotions on functional connectivity during subsequent resting state: a wavelet correlation approach. *Neuroimage* **54**, 2481-2491, (2011).
- 15 Benoit, O., Daurat, A. & Prado, J. Slow (0.7-2 Hz) and fast (2-4 Hz) delta components are differently correlated to theta, alpha and beta frequency bands during NREM sleep. *Clinical neurophysiology : official journal of the International Federation of Clinical Neurophysiology* **111**, 2103-2106, (2000).
- 16 Jobert, M., Escola, H., Poiseau, E. & Gaillard, P. Automatic analysis of sleep using two parameters based on principal component analysis of electroencephalography spectral data. *Biological cybernetics* **71**, 197-207, (1994).
- 17 Kuwahara, H. *et al.* Automatic real-time analysis of human sleep stages by an interval histogram method. *Electroencephalography and clinical neurophysiology* **70**, 220-229, (1988).
- 18 Steer, R. A., Ball, R., Ranieri, W. F. & Beck, A. T. Further evidence for the construct validity of the Beck depression Inventory-II with psychiatric outpatients. *Psychol. Rep.* **80**, 443-446, (1997).
- 19 Spielberger, C. D. Manual for the state-trait anxiety inventory. *Consulting Psychologists Palo Alto CA* (1983).

- 20 Johns, M. W. A new method for measuring daytime sleepiness: the Epworth sleepiness scale. *Sleep* **14**, 540-545, (1991).
- 21 Buysse, D. J., Reynolds, C. F., 3rd, Monk, T. H., Berman, S. R. & Kupfer, D. J. The Pittsburgh Sleep Quality Index: a new instrument for psychiatric practice and research. *Psychiatry research* **28**, 193-213, (1989).
- 22 Torrubia, R., Ávila, C., Moltó, J. & Caseras, X. The Sensitivity to Punishment and Sensitivity to Reward Questionnaire (SPSRQ) as a measure of Gray's anxiety and impulsivity dimensions. *Personality and Individual Differences* **31**, 837-862, (2001).
- 23 Schmidt, R. E., Gay, P. & Van der Linden, M. Facets of impulsivity are differentially linked to insomnia: evidence from an exploratory study. *Behavioral sleep medicine* **6**, 178-192, (2008).
- 24 Van der Linden, M. *et al.* A French Adaptation of the UPPS Impulsive Behavior Scale: Confirmatory Factor Analysis in a Sample of Undergraduate Students. *European Journal of Psychological Assessment* **22**, 38-42, (2006).

Reviewer #1 (Remarks to the Author):

The authors have satisfactorily answered all of my comments.

Reviewer #2 (Remarks to the Author):

This is an excellent example of a constructive review process and a revision that substantially strengthens the paper. The reviewers collectively made many helpful suggestions and the reviewers did an very thorough job addressing them. I have no further comments on this paper.

Reviewer #3 (Remarks to the Author):

General comment:

I thank the authors for their thorough responses to my comments and those of other reviewers. While the manuscript has been improved regarding several key aspects and while it offers a better demonstration that reward may guide memory reactivation during sleep, I still have important issues with the analyses. As similar issues have been raised by Reviewer 1, I think the authors should address them in full.

Comment #1:

As pointed out in my first review, the authors trained a classifier on 5 states but they seem interested in just two (Reward and No-Reward) of these 5 states (1 other serving as a control: Rest). The two remaining states (Pre-Face and Pre-Maze) did not seem part of the study's rationale ("We also included two states of no interest" l. 6). As these states also include very little data (1.96% of the total data used for the training of the classifier), it is very unclear what they were used at all. It is a major oddity of this paper and all the more surprising since it seems rather straightforward to fix it.

The justification of the authors ("We did this precisely to avoid that the classifier may be artificially biased towards one specific state") does not hold as it is precisely what the classifier did (see. Fig. S1 with the two additional states being predominantly classified as "Rest").

In my opinion, there is one clear alternative here:

- Either adding/removing these two states do not change the results and therefore, without any rationale to include them, the paper would be clearer without them.
- Or adding/removing these two states change the result and this must be reported and interpreted as there is currently no explanation for why it would be the case.

Comment #2:

I thank the authors for clarifying their statistical analyses. When performing the new repeated measure ANOVA, I strongly think the authors should keep the predictor of Game (which is, if I understood the design correctly, a factor that is crossed with Reward/NoReward between-subject). According to the authors' hypotheses, they should have an effect of Reward/NoReward and no effect of the type of Game. This contrast is important to conclude that the reactivation of the memory was really about the game valence. Similarly, since they now show individual dots in Figure 3, they could use different symbols for the Maze and Face Games (e.g. diamonds and circles). This will allow readers to visualize whether the increase in activation for the rewarded game was present for both games or favored one type of game. Currently, this effect is not analyzed or displayed and, yet, it is a central element of the design.

Comment #3:

I thank the Authors for the new Figure 4. In the panel A, can they show the individual masks for "N3: Face > Rest", "Wake: Face > Rest", "N3: Maze > Rest" and "Wake: Maze > Rest"? Currently they seem to show only the intersection between (i) "N3: Face > Rest" and "Wake: Face > Rest", and (ii) "N3: Maze > Rest" and "Wake: Maze > Rest". This is not ideal since, for example, knowing which brain regions show up in the "N3: Face > Rest" but not in "Wake: Face > Rest" is of interest.

Reviewer #1

Comment #1

The authors have satisfactorily answered all of my comments.

Response #1

We thank the reviewer for the validation of our replies.

Reviewer #2

Comment #1

This is an excellent example of a constructive review process and a revision that substantially strengthens the paper. The reviewers collectively made many helpful suggestions and the reviewers did an very thorough job addressing them. I have no further comments on this paper.

Response #1

We are grateful to the reviewer for the very positive comment about the revision of our manuscript. We also take the opportunity to thank this reviewer and the other reviewers for taking the time to read and so carefully comment on the different aspects of our paper. Their comments allowed us to improve considerably the presentation of the results of our study, and thereby to better demonstrate that reward biases spontaneous neural reactivation during sleep.

Reviewer #3

General comment:

I thank the authors for their thorough responses to my comments and those of other reviewers. While the manuscript has been improved regarding several key aspects and while it offers a better demonstration that reward may guide memory reactivation during sleep, I still have important issues with the analyses. As similar issues have been raised by Reviewer 1, I think the authors should address them in full.

We are happy that the reviewer was pleased with our previous revision. We address his/her residual concerns below.

Comment #1:

As pointed out in my first review, the authors trained a classifier on 5 states but they seem interested in just two (Reward and No-Reward) of these 5 states (1 other serving as a control: Rest). The two remaining states (Pre-Face and Pre-Maze) did not seem part of the study's rationale ("We also included two states of no interest" l. 6). As these states also include very little data (1.96% of the total data used for the training of the classifier), it is very unclear what they were used at all. It is a major oddity of this paper and all the more surprising since it seems rather straightforward to fix it.

The justification of the authors (“We did this precisely to avoid that the classifier may be artificially biased towards one specific state”) does not hold as it is precisely what the classifier did (see. Fig. S1 with the two additional states being predominantly classified as “Rest”).

In my opinion, there is one clear alternative here:

- Either adding/removing these two states do not change the results and therefore, without any rationale to include them, the paper would be clearer without them.
- Or adding/removing these two states change the result and this must be reported and interpreted as there is currently no explanation for why it would be the case.

Response #1

As requested by the reviewer, we ran new analyses, now excluding the pre-Reward and pre-NoReward states. Thus, the classifier was trained on only 3 states, including Face, Maze and Rest states. The figure below shows the classification of the fMRI data from the game session when training the classifier on the initial 5 states (Fig. R1A and B) and on only 3 states (Fig. R1C and D). The confusion matrices of the cross-validation (leave-one-out procedure) shared a similar general pattern but, and as expected, the classifier was slightly less efficient with 3 states, i.e. when excluding the pre-Face and pre-Maze conditions, than with the initial 5 states (Fig. R1E). Specifically, a repeated-measures ANOVA on the shared 3 states of interest with Method (5 or 3 states) and Game-related State (Face, Maze, Rest) as within-subjects factors revealed a main effect of Method ($F(1,17)$, 5.9, $p=0.026$) due to higher decoding accuracy for 5 compared to 3 states, a main effect of Game-related State ($F(2,34)=18.9$, $p<0.001$), higher accuracy for Rest compared to Face and Maze states, and no interaction ($F(2,34)=3.6$, $p=0.37$).

Figure R1. Task-related decoding accuracy ($N=18$). (A) Confusion matrix and (B) individual decoding accuracy measures for 5 states. (C) Confusion matrix and (D) individual decoding accuracy measures for 3 states. (E) Summary measures for the states Face, Maze, Rest when using 5 and 3 states.

We then performed all the analyses reported in the original manuscript, but now using the output of the classifier on 3 states. For the decoding of the data from the Sleep session, a repeated-measures ANOVA on likelihood measures with Brain state (Reward, No-Reward, Rest) and Sleep stage (wake, N1, N2, N3) as within-subjects factors showed main effects of Brain state ($F_{(2,24)}=86.3$, $P < 0.001$) and an interaction between Brain state and Sleep stage ($F_{(6,72)} = 13.4$, $P < 0.001$) (Fig. R2A). Post-hoc planned comparisons first replicated the original results, showing that the Rest state predominated over the Reward and No-Reward states during wakefulness (Rest vs. Reward: $F_{(1,12)}=185.1$, $p < 0.001$, Rest vs. No-Reward: $F_{(1,12)}=389.2$, $p < 0.001$), while it was progressively less represented from light to deeper sleep stages (main effect of sleep for the Rest state: $F_{(1,12)}=22.14$, $p < 0.001$). Conversely, both game-related brain states did not differ between Reward versus No-Reward states from wakefulness to N2 (Reward vs. No-Reward for Wake: $F_{(1,12)}=2.1$, $p=0.18$, N1: $F_{(1,12)}=0.002$, $p=0.96$, N2: $F_{(1,12)}=0.22$, $p=0.64$). By contrast, and as demonstrated in the original analysis, the Reward state differed from the No-Reward state during N3 sleep ($F_{(1,12)}=15.65$, $p=0.002$, Fig. R2B).

We also extracted the power values for the relevant frequency bands from the EEG data over the whole sleep session, and correlated these values with the time course of likelihood for the 3 different states detected in the fMRI data from each participant. The resulting correlation values entered an ANOVA with Brain state (Reward, No-Reward, Rest) and Frequency band (low delta, 1-2 Hz; high delta, 2-4 Hz; theta, 4-7 Hz; alpha, 8-10 Hz; sigma, 12-14 Hz; beta, 15-25 Hz) as within-subjects factor. We observed a main effect of Brain state ($F_{(2,24)}=7.35$, $p=0.003$), a main effect of Frequency band

($F_{(5,60)}=9.19$, $p<0.001$), and an interaction between both factors ($F_{(10,120)}=14.77$, $p<0.001$). More specifically, a direct comparison between Reward and No-Reward states (planned comparison) showed a significant difference ($F_{(1,12)}=5.84$, $p=0.032$) due to the strong positive correlation between low delta activity and the strength of the reactivation of the brain state associated with the successful game (Fig. R2C). No such interaction was found when exploring correlations of game-related states with other frequency bands.

Figure R2. Classification results with only 3 states (N=13). (A) Mean likelihood of Reward, No-Reward, Rest brain states for each sleep stage, showing increased reactivation of the brain state associated with the rewarded game during N3. (B) Individual data for the Reward and No-Reward state during N3 sleep. (C) Correlation between time course of likelihood and low delta power (1-2 Hz) for the Reward and No-Reward states. Error bars represent SEM; dots represent individual data points.

Next, we ran the fMRI analysis with likelihood values for the task-specific brain states (face, maze) for each fMRI volume during N3 sleep as regressors in a whole-brain regression analysis, independently of the reward status and computed a conjunction analysis, as described in the previous version of the manuscript. We observed a similar set of brain regions activated as when using 5 brain states in the classification (see original Figure 4). We also extracted the beta values from each network and correlated the mean beta values with memory performance (Fig R3B). Significant correlations were obtained between memory performance and fMRI signal during N3, thus replicating the original results.

Figure R3. Game-specific reactivation during N3 and correlation with subsequent memory using 3 game-related states (N=13). (A) In red, brain regions more activated for Face than Rest brain state during N3 sleep, and also recruited during the execution of the task at wake (blocks of Face vs. Rest; conjunction analysis). In blue, brain regions more activated for Maze than Rest brain state during N3 sleep, and also recruited during the maze game. In purple, overlap between those face and maze reactivation networks during sleep. (B) Correlation between activity in the face network (red regions in A) during N3 and memory performance for the face game. (C) Correlation between activity of the maze network (blue regions in A) during N3 and memory for the maze game. Grey dots for participants who won the face game; black dots for participants who won the maze game. FFA: fusiform face area; paraH: parahippocampus; OFA: occipital face area. N=13.

Finally, we assessed the correlation between the time courses of activity from bilateral hippocampus and bilateral VTA and the 3 different brain states decoded during the sleep session (Reward, NoReward and Rest), and performed two ANOVAs on the rho values, with the 3 states as a within-subjects factor (Fig. R4). For the hippocampus, we observed a main effect of State ($F_{(2,24)}=98.1$, $p<0.001$). Post-hoc analyses revealed that the time course of hippocampal activity correlated significantly more with the Reward state than with the other states (vs. No-Reward state: $F_{(1,12)}=12.8$, $p=0.004$; vs. Rest state: $F_{(1,12)}=100.9$, $p<0.001$). Similarly, for the VTA, we observed a main effect of State ($F_{(2,24)}=25.1$, $p<0.001$), due to the time course of VTA activity correlating more with the Reward state than the other ones (vs. No-Reward state: $F_{(1,12)}=10.2$, $p=0.007$; vs. Rest state: $F_{(1,12)}=28.3$, $p<0.001$).

Figure R4. Reward-related reactivation in hippocampal and ventral tegmental area (VTA) during sleep, computed for the results of the classifier on only 3 game-related states (N=13). (A) Anatomical masks used as regions of interest (in green). (B) Mean correlation values for all participants between the time courses of the regions of interest and the different brain states. Asterisks indicate $P < 0.01$ for the Reward vs. No-Reward comparison.

In sum, the confusion matrix demonstrated that, despite the fact that the pre-Face and pre-Maze are of shorter duration and lower accuracy as compared to the other states, these periods are useful for the categorization of brain states during sleep. Theoretically, adding pre-game conditions could either improve or decrease the efficiency of the classifier. The additional analyses demonstrate that the pre-game periods actually accounted for some additional variance in the data. Importantly, the subsequent analyses performed using the output of the classifier trained on only 3 states replicated all the main results from the previous version of the manuscript. Based on these observations, we suggest that reporting the results with the 5 states (as in the original manuscript) would be more appropriate. If the Editor and/or Reviewer think that these additional results on the 3 states should be included as supplemental information, we would be delighted to do so.

Comment #2:

I thank the authors for clarifying their statistical analyses. When performing the new repeated measure ANOVA, I strongly think the authors should keep the predictor of Game (which is, if I understood the design correctly, a factor that is crossed with Reward/NoReward between-subject). According to the authors' hypotheses, they should have an effect of Reward/NoReward and no effect of the type of Game. This contrast is important to conclude that the reactivation of the memory was really about the game valence. Similarly, since they now show individual dots in Figure 3, they could use different symbols for the Maze and Face Games (e.g. diamonds and circles). This will allow readers to visualize whether the increase in activation for the rewarded game was present for both games or favored one type of game. Currently, this effect is not analyzed or displayed and, yet, it is a central element of the design.

Response #2

In the previous round of revision, we wanted to highlight the Reward (vs No-Reward) effect but it is true that, in this process, we removed all results showing that the pattern of results was independent

of the game won. To directly address the reviewer’s point, we ran a new repeated-measures ANOVA on likelihood measures with Brain state (Reward, No-Reward, Rest, pre-R, pre-NR) and Sleep stage (wake, N1, N2, N3) as within-subjects and Game Won as between-subjects factor. We now report the results from this new analysis in the Results section of the revised version of the manuscript (page 7):

“To ensure that the type of game won (Face or Maze) did not affect the main results, we computed a new ANOVA with Brain state and Sleep stage as within-subjects factor and Game Won as between-subjects factor. We observed no main effect of Game Won ($F(1,11)=0.25$, $p=0.63$), no interaction between Game Won and Brain state ($F(4,44)=0.27$, $p=0.89$) and interaction between Game Won and Sleep stages ($F(3,33) = 0.46$, $p=0.71$), thus further suggesting that neural reactivation was enhanced by the associated reward value, independently of the type of game.”

We also modified the original Figure 3 (replicated below) to graphically distinguish data points from individual who won the fae game (dots) from those who won the maze win-group (diamonds). Visually, these individual measures appeared to be relatively homogeneously distributed in each condition, as would be expected from the ANOVA results reported above.

Figure 3. Classification results (N=13). (A) Mean likelihood of Reward, No-Reward, Rest, pre-Reward (pre-R) and pre-No-Reward (pre-NR) brain states for each sleep stage, showing increased reactivation of the brain state associated with the rewarded game during N3. (B) Individual data for the Reward and No-Reward state during N3 sleep. (C) Correlation between time course of likelihood and low delta power (1-2 Hz) for the Reward and No-Reward states. Error bars represent SEM; dots represent individual values for those participants won the face game and diamonds correspond to data from participants who won the maze game.

Comment #3:

I thank the Authors for the new Figure 4. In the panel A, can they show the individual masks for “N3: Face > Rest”, “Wake: Face > Rest”, “N3: Maze > Rest” and “Wake: Maze > Rest”? Currently they seem to show only the intersection between (i) “N3: Face > Rest” and “Wake: Face > Rest”, and (ii) “N3: Maze > Rest” and “Wake: Maze > Rest”. This is not ideal since, for example, knowing which brain regions show up in the “N3: Face > Rest” but not in “Wake: Face > Rest” is of interest.

Response #3

We fully agree with the reviewer that the conjunction does not convey all the underlying information. Yet, because the conjunction between both contrasts directly tests for the relevant hypothesis (i.e. spontaneous reactivation = activation of brain networks activated during some prior task), we would still prefer to display the conjunction in Figure 4 in the main text. We now report the singular contrasts in a supplementary Figure S3 in the revised version of the manuscript (see below).

Game-specific reactivation during N3 (N=13). Upper panel shows the brain regions activated for the Face vs Rest conditions during wakefulness (red) and sleep (yellow), and overlapping activations (orange). Lower panel shows the brain regions activated for the Maze vs Rest conditions during wakefulness (blue) and sleep (light green), and overlapping activations (dark green).

Reviewer #3 (Remarks to the Author):

I thank the authors for addressing my last comments thoroughly. The authors were able to fully replicate their results using a 3-class classifier, which answers my concerns. If possible, it would be interesting to add these new analyses to the supplementary material.

In conclusion, this is an excellent paper and I would like to thank, once again, the authors for their engagement in the peer reviewing process. I wish also to congratulate them for what will surely be an impactful paper.